# Amnesia: A Stealthy Replay Attack on Continual Learning Dreams

**Ahmed Sharshar**                                                        *ahmed.sharshar@mbzuai.ac.ae*
*Department of Computer Vision*
*Mohamed bin Zayed University of Artificial Intelligence, AbuDhabi, UAE*

**Naveen Kumar Kummari**                                                  *naveen.kummari@mbzuai.ac.ae*
*Department of Machine Learning*
*Mohamed bin Zayed University of Artificial Intelligence, AbuDhabi, UAE*

**Mohsen Guizani**                                                        *mohsen.guizani@mbzuai.ac.ae*
*Department of Machine Learning*
*Mohamed bin Zayed University of Artificial Intelligence, AbuDhabi, UAE*

**Reviewed on OpenReview:** *https://openreview.net/forum?id=QSTg7zO6GH*

## Abstract

Continual learning (CL) models rely on experience replay to mitigate catastrophic forgetting, yet their robustness to replay sampling interference is largely unexplored. Existing CL attacks mostly modify inputs or update pipelines (poisoning/backdoors) and lack explicit *auditable* constraints, limiting their realism. Here, *auditability* means that a monitor can verify compliance using sampler-visible telemetry, e.g., logged replay index/label statistics, by checking that the realized replay class histogram stays close to a nominal baseline and that the replay rate is unchanged (per-batch and/or over a rolling window). We study a limited-privilege insider controlling only the replay *index selection*, not pixels, labels, or model parameters, while staying within such auditable limits (e.g., queue priorities). We introduce **Amnesia**, a replay composition attack maximizing model degradation under two auditable budgets: a visibility budget $\delta$ bounding the TV/KL divergence from a nominal class histogram $p_0$, and a mass budget $f$ fixing the replay rate. Amnesia uses a two-step procedure: (i) compute lightweight class utilities (e.g., EMA loss/confidence) to tilt $p_0$ toward harmful classes; (ii) project the tilt back into the $\delta$-ball using efficient KL (*exponential tilt*) or TV (*balanced mass redistribution*) optimizers. A windowed scheduler enforces rolling audits. Across challenging CL benchmarks (Split CIFAR-10/100, CORe50, Tiny-ImageNet) and strong replay baselines (ER, ER-ACE, SCR, DER++), Amnesia consistently depresses final accuracy (ACC↓) and worsens backward transfer ($-$BWT ↑). The KL variant achieves high impact while remaining largely undetected by audits, as confirmed empirically under multiple audit schemes (per-batch and rolling-window checks), whereas the TV variant is more damaging but more easily detected, especially under tight per-class constraints. These results expose *index-only* replay control as a practical, auditable threat surface in CL systems and establish a principled impact-visibility-budget trade-off. Code is available anonymously via GitHub.

## 1 Introduction

Continual learning (CL) aims to sequentially adapt to evolving data while avoiding catastrophic forgetting (Chaudhry et al., 2019a; Kirkpatrick et al., 2017). Managing the plasticity-stability trade-off (Lange et al., 2021) underpins improved forward and backward transfer (FWT/BWT) (Lopez-Paz & Ranzato, 2017; Lange et al., 2021). This capability is crucial for non-stationary applications such as robotics (Ye & Bors, 2025)

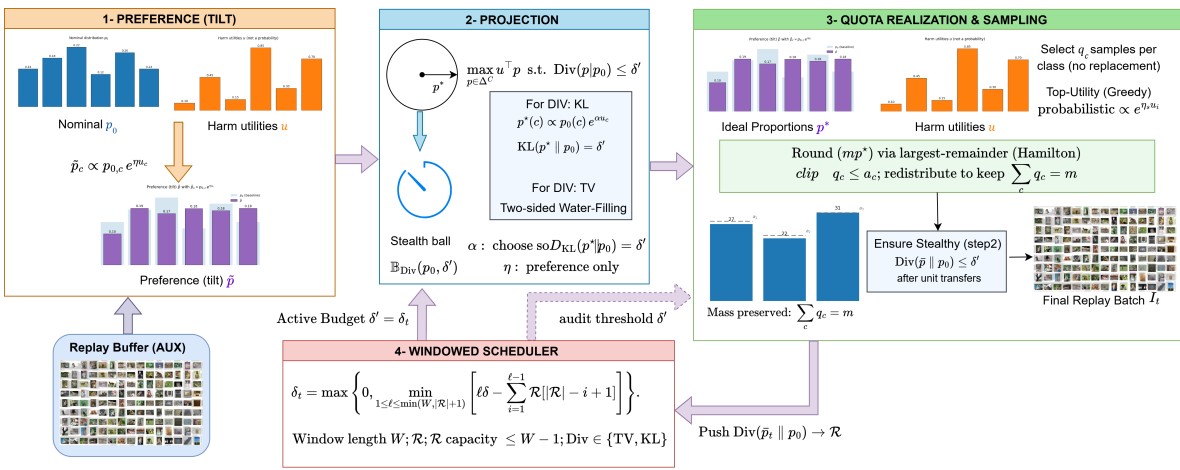

Figure 1: **Amnesia attack overview.** Four-stage pipeline: (1) *Preference*: tilt the nominal class histogram $p_0$ using harm utilities $u$ to obtain $\tilde{p}$. (2) *Projection*: map into the stealth (divergence) ball $\mathbb{B}_{\text{Div}}(p_0, \delta')$ (total variation / Kullback-Leibler; TV/KL) to get $p^\star$. (3) *Quota & sampling*: round $mp^\star$ to integer quotas $q$, clip/audit to keep $\text{Div}(\bar{p} \| p_0) \le \delta'$, then sample the batch indices $I_t$. (4) *Windowed scheduler*: a ring buffer $\mathcal{R}$ sets the active budget over a rolling window $W$.

and spans task, class, and data-incremental regimes (van de Ven et al., 2022) (e.g., distinct tasks, expanding label sets, or distribution shift under a fixed label set). Replay methods interleave a small memory of past examples with current data to mitigate forgetting (van de Ven et al., 2022), often outperforming regularization alternatives (Chaudhry et al., 2019b; Lopez-Paz & Ranzato, 2017) and proving effective in reinforcement learning (Schaul et al., 2016). Yet the *robustness* of CL pipelines to malicious interference remains underexplored.

In deployed CL systems, replay is often implemented as a sampler/data-service that is instrumented for reliability and compliance: it logs replay index sets, per-batch label counts, queue/priority metadata (e.g., in prioritized replay), and the replay rate (resource usage). Such signals enable lightweight *audits* that verify simple invariants without inspecting model parameters or the live stream, e.g., that the replay keep fraction stays fixed and that the replay label histogram remains close to a nominal baseline derived from the buffer. We call constraints of this form *auditable budgets* because they are (i) defined on telemetry that is naturally logged in production pipelines and (ii) verifiable post hoc (per batch and/or over rolling windows) via automated compliance checks. We address the above robustness gap by formalizing these auditable budgets and proposing a *sampler-level*, divergence-constrained replay composition attack operating under explicit visibility and mass limits—to the best of our knowledge, the first to target replay *index selection* under such audit-aligned constraints.

A growing body of work attacks CL via *poisoning* (distribution perturbations) and *backdoors* (input-space triggers). Poisoning aims to degrade overall performance, whereas backdoors implant dormant triggers for targeted failures. For example, *BrainWash* shows that poisoning the *current* task can erase knowledge of *past* tasks (Abbasi et al., 2024); biased synthetic samples can subvert generative replay (Kang et al., 2023); and task-specific poisoning degrades regularization-based CL by exploiting stability assumptions (Han et al., 2023). Backdoor variants such as *PACOL* (Umer et al., 2023) and persistent attacks across task sequences (Guo et al., 2025) leverage low-intensity triggers or temporally embedded patterns. However, these methods typically omit *practical* constraints that are central in monitored deployments (Steinhardt et al., 2017; Jagielski et al., 2018): (i) an *auditable visibility budget* bounding divergence ($\delta$) between the attacked replay histogram and a nominal baseline $p_0$ (as checked from selection logs), and (ii) a *mass budget* ($f$) limiting the replay rate (as checked from resource/throughput accounting). Such budgets are key for realistic, stealthy attacks (Namkoong & Duchi, 2016; Sinha et al., 2018).

We introduce **Amnesia**, a *divergence-constrained, sampler-level* replay composition attack (overview in Fig. 1). Rather than corrupting data, losses, or parameters, Amnesia biases *which indices* are drawn from the buffer (the "dreaming" stage), reflecting practical threats (e.g., a compromised index-selection service, flipped metadata priority flags, or tampered pseudorandom number generator (PRNG) seeds for prioritized replay (Schaul et al., 2016)). Importantly, the sampler need not query the model for real-time per-index losses: the utility signals it consumes can be lightweight, lagged telemetry (e.g., EMA loss/confidence) exported asynchronously by the training service and logged as buffer metadata. We pose the attack as:

$$\max_{p \in \Delta^C} \langle u, p \rangle \quad \text{s.t.} \quad \mathrm{Div}(p\|p_0) \leq \delta, \ \ \mathrm{Div} \in \{\mathrm{TV, KL}\}, \tag{1}$$

where $C$ is the number of classes, $u$ is the attacker's per-class harm utility, and $\delta$ is the maximum allowed total variation (TV) distance or Kullback-Leibler (KL) divergence from $p_0$ (the visibility budget). This is coupled with a mass budget: a keep fraction $f$ that fixes the total replay mass $m = \lfloor f\, n_{\mathrm{aux}} \rfloor$, where $n_{\mathrm{aux}}$ is the size of the auxiliary replay buffer. The optimizer's solution, $p^\star$, is then realized as integer quotas $q$ (with $\sum_c q_c = m$); the resulting normalized histogram $\bar{p} = q/m$ is what must pass the audit. The optimizer is agnostic to how $u$ is estimated (e.g., class-wise exponential moving average (EMA) losses, misclassification sensitivity, or forgetting proxies). *Remark:* if $u$ is uninformative (e.g., constant across classes) or $\delta = 0$, then $p^\star = p_0$ and the attack reduces to the nominal replay sampler.

This formulation yields efficient, exact optimizers: a KL *single-tilt* (exponential tilt with a one-dimensional search for the budget-saturating multiplier) and a TV *two-sided water-filling* (balanced mass redistribution between low- and high-utility classes). The attack is designed to maximize forgetting on past tasks while preserving current-task accuracy for stealth, and we evaluate it along three axes: *impact* (backward transfer, BWT; forgetting, FGT (average accuracy drop on past tasks)), *visibility* ($\delta$), and *budget* ($f$). Under fixed replay mass $m$, shifting replay probability toward attacker-designated classes necessarily reduces rehearsal for other classes, which is the basic mechanism by which forgetting can be amplified.

Conceptually, Amnesia tilts sampling toward attacker-designated harmful classes while enforcing $\delta$ and $f$. The nominal histogram $p_0$ may be the instantaneous empirical distribution or a moving average, enabling audits per batch or over a rolling window $W$. A *windowed scheduler* based on a ring buffer $\mathcal{R}$ tightens the per-step budget to an active $\delta_t \leq \delta$ so that any window of length $L \leq W$ stays within the global radius while meeting integer quotas. This preference-under-constraints view connects replay sampling to our baselines, **PO** (Preference-Only tilt) and **PrO** (Projection-Only fairness/coverage via KL/TV projection around $p_0$)— but *inverts the goal under the same audit-aligned constraints*: rather than *defending* by minimizing worst-case risk within a divergence ball or enforcing coverage, we *attack* by maximizing targeted harm while remaining compliant with logging-based replay audits (rate and histogram checks). By construction, our optimizer *weakly dominates* any other feasible sampler-level attack (including simple budget-aware greedy heuristics) at an equivalent $(f, \delta)$, and *strictly dominates* when $u$ is non-uniform and $\delta > 0$. In summary, our contributions are as follows:

- **Amnesia:** a practical *sampler-level* replay composition attack that steers class proportions by manipulating replay index selection, leaving pixels, loss computations, and model parameters untouched.

- **Principled formulation:** (i) *Attack formulation under auditable budgets:* an explicit divergence-constrained optimization over replay proportions $p \in \Delta^C$ with a *visibility budget* $\mathrm{Div}(p\|p_0) \leq \delta$ (TV/KL distance from the nominal replay histogram) and a *mass budget* $f$ fixing the replay rate $m = \lfloor f\, n_{\mathrm{aux}} \rfloor$; (ii) *Audit realization on what monitors can check:* a sampler-side realization pipeline that converts $p^\star$ into integer quotas $q$, handles availability constraints, and performs post-rounding *audit-and-fix* so the *realized* replay histogram $\bar{p} = q/m$ satisfies $\mathrm{Div}(\bar{p}\|p_0) \leq \delta'$, with a windowed scheduler enforcing rolling-window audits.

- **Comprehensive evaluation:** experiments on Split CIFAR-10/100, CORe50, and Tiny-ImageNet, against strong replay baselines (ER, ER-ACE, SCR, DER++), systematically analyzing the *impact–visibility–budget* trade-off, damage-per-budget, and attack detectability under multiple audit schemes.

## 2 Related Work

In continual learning (CL), models learn from streams of tasks or shifting distributions while mitigating catastrophic forgetting (van de Ven et al., 2022; Kirkpatrick et al., 2017). As reviewed in §1, canonical settings are *task-incremental* (task identity known at test), *domain-incremental* (also known as (a.k.a.) data-incremental in some works; shared label space with distribution shift), and *class-incremental* (recognition over all seen classes without task labels) (van de Ven et al., 2022; De Lange et al., 2022; Cossu et al., 2022). CL is also studied in *online, single-pass* streams, especially with many small tasks (Chaudhry et al., 2019b). From the evolving accuracy matrix, *Average Accuracy (ACC)*, *Backward Transfer (BWT)*, and *Forward Transfer (FWT)* (typically vs. random initialization) capture final performance, forgetting, and positive transfer (Díaz-Rodríguez et al., 2018; Hou et al., 2024). Method families include *memory-based (replay)*, *regularization-based*, and *architectural/parameter isolation*, with replay dominant in vision; small buffers curb forgetting and are maintained via *reservoir sampling* or *ring-buffer* updates (De Lange et al., 2022; Chaudhry et al., 2019b; Mai & contributors, 2020). In practice, replay batches are formed by a sampler module (often implementing class-balancing or prioritized replay), which exposes a natural control point for attacks that manipulate *which* buffer indices are rehearsed. Strong baselines such as *GDumb* highlight buffer composition, while distillation-enhanced replay (e.g., *DER/DER++*) stabilizes predictions; regularization (e.g., elastic constraints) and associated analyses illuminate forgetting (Prabhu et al., 2020; Buzzega et al., 2020; Kirkpatrick et al., 2017; Huszár, 2018; Shen et al., 2023). The objective is high ACC, non-negative BWT, and positive FWT under realistic memory/compute budgets (Díaz-Rodríguez et al., 2018; De Lange et al., 2022).

Sequential training opens attack surfaces beyond static settings: (i) **poisoning**: insert or modify stream samples to degrade retention or bias behavior (targeted erasure (Li & Ditzler, 2022); BRAINWASH uses norm-bounded current-task perturbations and particularly harms Elastic Weight Consolidation (EWC) (Abbasi et al., 2024; Kirkpatrick et al., 2017; Huszár, 2018)); small budgets suffice via sequential amplification (Guo et al., 2025), and replay can repeatedly rehearse poisons (Lopez-Paz & Ranzato, 2017; Chaudhry et al., 2019a); (ii) **backdoors (Trojans)** persisting across tasks, including controllable backdoors effective against regularization- and replay-based learners (Gao & Liu, 2025; 2024), and *Persistent Backdoor Attacks*: *Blind Task Backdoor* (BTB, per-task) and *Latent Task Backdoor* (LTB, single-task "sleeper" activated when a future target class appears), achieving high attack success with minimal clean-accuracy drop (Guo et al., 2025); (iii) **test-time evasion** where standard adversarial examples remain effective, motivating *Continual Adversarial Defense (CAD)* and *Retrospective Adversarial Replay (RAR)* (Wang et al., 2023; Kumari et al., 2022); and (iv) **distribution/scheduling attacks** that adversarially order tasks to exacerbate interference and forgetting, exploiting order sensitivity and the difficulty of the class-incremental regime (van de Ven et al., 2022; De Lange et al., 2022; Cossu et al., 2022). These threats imply that robust CL must handle sequentially amplified poisoning, long-lived backdoors, non-stationary adversaries, and adversarial curricula.

We systematize these threats along three axes: **Attack Budget (AB)**, **Attack Visibility (AV)**, and **Attack Impact (AI)**. *AB*: attacker control over the stream or training pipeline (poisoned fraction, task access, perturbation magnitude); sequential training can amplify small budgets, e.g., targeted erasure (Li & Ditzler, 2022), ∼4% poisoning in backdoors (Guo et al., 2025), or norm-bounded task-$t$ perturbations in BRAINWASH (Abbasi et al., 2024). Temporal access matters: *LTB* needs single-task access, whereas *BTB* assumes per-task intervention (Guo et al., 2025); controllable backdoors in class-incremental CL succeed against both regularization- and replay-based learners (Gao & Liu, 2025; 2024). Replay revisitation can exacerbate small-budget poisoning (Lopez-Paz & Ranzato, 2017; Chaudhry et al., 2019a). *AV*: stealth, often preserving current-task accuracy or using rare triggers; BRAINWASH contrasts "reckless" vs. "cautious" trade-offs (Abbasi et al., 2024); persistent backdoors aim for high clean accuracy with long-term success (Guo et al., 2025; Gao & Liu, 2025). *AI*: damage and persistence, backdoors via attack success across subsequent tasks (Guo et al., 2025; Gao & Liu, 2025; 2024); poisoning-induced forgetting via negative BWT and ACC drops (Díaz-Rodríguez et al., 2018), with certain regularizers being notably vulnerable (Kirkpatrick et al., 2017; Huszár, 2018). Despite their centrality, AB/AV/AI are rarely reported jointly or standardized, motivating unified evaluation.

# 3 Methodology

We study a *sampler-level composition attack* for continual learning (CL) with experience replay. The adversary controls only the replay *index set* and seeks to maximize forgetting *(measured as the decline in past-task accuracy)*. The attack has two stages: (1) **Preference**: use lightweight utility signals (e.g., loss, confidence, or other logged scores) to *tilt* the nominal class histogram ($p_0$), producing a harm-biased target mix; and (2) **Projection**: map this mix into the auditable *stealth ball* ($\mathbb{B}_{\text{Div}}(p_0, \delta)$) with $\text{Div} \in \{\text{TV}, \text{KL}\}$. Projection guarantees each replay batch remains within divergence ($\delta$) of ($p_0$), keeping batch-level telemetry (the quantities visible to audits) plausibly benign. Unlike harm-agnostic baselines (e.g., **PO** (Preference-Only)) or fairness/coverage quota schemes (e.g., **PrO** (Projection-Only)), we *explicitly* insert the harm-based preference *before* projection, optimizing impact while staying within the same auditable constraints (Kumar et al., 2024).

Crucially, the "tilt-then-project" step is *computationally lightweight*: for **TV**, the exact optimizer is a greedy two-sided water-filling procedure (Algorithm 2 (ProjectTV)); for **KL**, it reduces to a single-parameter exponential tilt with a monotone 1-D root search (Algorithm 2(ProjectKL)). If utilities are uninformative (e.g., $u$ is constant) or if the active budget is $\delta' = 0$, then projection returns $p^\star = p_0$ and the attack has no effect.

## 3.1 Preliminaries & Notation

We consider a continual learner with a labeled replay buffer $\mathbf{AUX} = \mathcal{A} = \{(x_i, y_i)\}_{i=1}^{n_{\text{aux}}}$. In class-incremental CL, the set of seen classes grows over time; we let $C_t$ denote the number of classes observed up to step $t$, interpret labels as $y_i \in [C_t]$, and represent all class histograms (e.g., $p_0$, $p^\star$, $\bar{p}_t$) as elements of $\Delta^{C_t}$. When new classes appear, we conceptually append new coordinates and recompute $p_0$ from the current buffer histogram; the divergence audit $\text{Div}(\bar{p}_t \| p_0)$ is then evaluated in this expanded space. While our attack applies to evolving buffers (e.g., sliding windows), we assume $\mathcal{A}$ is a fixed reservoir for notation simplicity. The nominal (audited) class histogram is $p_0 \in \Delta^{C_t}$, where $\Delta^{C_t} = \{p \in \mathbb{R}_+^{C_t} : \sum_c p_c = 1\}$ is the probability simplex; we assume $p_{0,c} > 0$ for all $c$ (or use standard smoothing) so that $\text{KL}(p \| p_0)$ is finite whenever $p_c > 0$. For brevity, when the step $t$ is clear we sometimes write $C$ as shorthand for the current $C_t$.

At each training step $t$, the attacker must select a replay batch of a fixed *mass* (size) $m = \lfloor f \, n_{\text{aux}} \rfloor$, determined by a public keep fraction $f$. This selection is represented by an index set $I_t$, which corresponds to integer per-class quotas $q_t \in \mathbb{N}_0^C$ (with $\mathbb{N}_0$ for non-negative integers) where $\sum_c (q_t)_c = m$, and realizes the per-batch histogram $\bar{p}_t := q_t / m$. Let $a_c$ be the available samples per class; quotas are clipped ($q_{t,c} \leq a_c$) and redistributed to preserve $\sum_c q_{t,c} = m$.

**Harm utilities and how they are obtained.** The attacker's objective is to maximize harm, quantified by per-class utilities $u = (u_1, \ldots, u_C) \in \mathbb{R}^C$. Optional per-sample utilities $\tilde{u}_i$ (e.g., loss/confidence) are logged as *scalar metadata* alongside AUX items, and class utilities are maintained as EMAs, $u_c^{(t)} = \rho \, u_c^{(t-1)} + (1 - \rho) \, \text{Agg}\{\tilde{u}_i : y_i = c\}$. Importantly, the sampler need not query the *current* model online: $\tilde{u}_i$ can be produced by the training service when an AUX sample is replayed (then stored), refreshed periodically (batch jobs), or replaced by simpler proxies (e.g., age, misclassification counters). If these signals are stale/noisy, the attack degrades gracefully; if $u$ is effectively constant, the optimizer returns $p^\star = p_0$.

We work with divergences $\text{Div} \in \{\text{TV}, \text{KL}\}$, defined as $\text{TV}(p \| p_0) = \frac{1}{2} \| p - p_0 \|_1$ and $\text{KL}(p \| p_0) = \sum_c p_c \log\left(\frac{p_c}{p_{0,c}}\right)$. A *stealth radius* $\delta > 0$ defines the *stealth ball* $\mathbb{B}_{\text{Div}}(p_0, \delta) := \{p \in \Delta^C : \text{Div}(p \| p_0) \leq \delta\}$. For systems with windowed auditing (length $W$), the window-average histogram is $\hat{p}_{t-L+1:t} := \frac{1}{L} \sum_{s=t-L+1}^t \bar{p}_s$. A ring buffer $\mathcal{R}$ tracks past divergences, and the attacker computes an *online tightened budget* $\delta_t \leq \delta$; we use $\delta'$ to denote the active budget (either $\delta$ or $\delta_t$) at step $t$. Throughout, we treat $p \in \Delta^C$ as a probability vector (sum = 1); the keep fraction $f$ appears only via the batch mass $m = \lfloor f \, n_{\text{aux}} \rfloor$ when realizing $p^\star$ as integer quotas, and all divergences are evaluated on normalized histograms (e.g., $\bar{p}_t = q_t / m$). Finally, class-level preference strength is denoted by $\eta > 0$ and within-class (intra-class) selection by a temperature $\eta_s > 0$.

### 3.2 Threat Model & Attack Surface

We consider a **grey-box** insider who controls only the replay sampler and aims to maximize forgetting under fixed replay mass $f$ and an audited stealth radius $\delta$; Table 1 summarizes capabilities and limitations.

We assume the auditor's nominal baseline $p_0$ is computed as a deterministic function of sampler-visible telemetry (e.g., the replay-buffer label histogram at sampling time, or a moving average thereof), so it can be reproduced by the sampler. If the attacker instead observes only a lagged/noisy estimate $\hat{p}_0$, baseline mismatch effectively reduces the usable visibility margin; Appendix B.2 (Table 9) quantifies the resulting impact–stealth trade-off by varying the lag between $\hat{p}_0$ and $p_0$.

**Scope.** Our threat model assumes *replay-based* CL with a stored buffer and a distinct replay sampler that selects indices $I_t$. Accordingly, Amnesia does *not* directly apply to *rehearsal-free* CL methods (e.g., prompt-based approaches such as OVOR Huang et al. (2024) or CODA-Prompt Smith et al. (2023)) because they do not maintain a replay buffer or replay-index selection surface. Conversely, the attack surface is orthogonal to whether the learner fine-tunes the full backbone or updates only a parameter-efficient module (e.g., head/adapters/LoRA/prompts): whenever replay is used, changing the sampled replay indices changes the training signal seen by whatever parameters are trainable. Extensions to other CL pipelines that sample "past information" (e.g., prototype selection or class-conditional generative replay) are conceptually related but outside our experimental scope. More broadly, this suggests potential relevance beyond replay-based CL: future non-replay settings, including continual learning for language models, may expose analogous low-privilege attack surfaces wherever a mechanism selects or composes past information (e.g., retrieval/memory selection, prompt/example selection, or routing), even without an explicit replay buffer; we leave direct study of such settings to future work.

Table 1: Threat model: insider control of the replay sampler with constrained budgets and audited visibility.

| Capabilities | Attacker Limitations |
|---|---|
| Direct control of the replay sampler: set per-class quotas/probabilities; realize index sets $I_t$ with total mass $m$. | No access to model internals (weights, gradients, optimizer). |
| Read-only visibility of AUX-buffer labels and *stored* utility scores ($\tilde{u}_i$, class-wise $u_c$); signals may be lagged EMAs or periodically refreshed logs. | No read or write access to the live task stream (pixels, labels) and no requirement to query real-time per-index losses from the trainer (utilities are treated as logged metadata). |
| Temporal planning: schedule compositions and precommit plans for the next $N$ batches. | Audited controls: fixed keep fraction $f$ and stealth radius $\delta$; TV/KL checks on label histograms vs. baseline $p_0$. |
| | Active monitoring: auditors track per-batch/window histograms/divergences and cross-check sampler code/config and selection logs. |

This attack surface is realistic in modern MLOps pipelines. Integration points include sampler plugins that intercept batch requests (e.g., `get_replay_batch()`), configuration toggles for "balanced replay" that route to a quota module, and data-ops jobs that precompute a selection plan for future windows. Legitimate controls such as *class-balancing*, *prioritized replay*, and *curriculum sampling* are commonly exposed via standard **PyTorch** sampler/data-loader hooks and the same utility telemetry, making this insider surface viable. The attacker can either persistently write malicious index sets or have the sampler execute a precomputed plan.

### 3.3 Preference–Projected Replay Attack

This subsection details the sampler-side routine as shown in Fig. 1. It runs once per training step, manipulates only replay *indices*, and leaves model weights/gradients/loss unchanged. At each step we solve the class-level program

$$\max_{p \in \Delta^C} u^\top p \quad \text{s.t.} \quad \text{Div}(p \| p_0) \leq \delta', \quad \text{Div} \in \{\text{TV}, \text{KL}\}, \tag{2}$$

where $u$ are harm utilities and $\delta'$ is the active stealth budget (either the static $\delta$ or the dynamic windowed budget $\delta_t$).

---

**Algorithm 1** Amnesia Replay Attack

---

**Require:** AUX buffer $\mathcal{A} = \{(x_i, y_i, \tilde{u}_i)\}_{i=1}^{n_{\text{aux}}}$; nominal histogram $p_0 \in \Delta^C$; keep fraction $f$; stealth radius $\delta$; window $W$; divergence $\text{Div} \in \{\text{KL}, \text{TV}\}$; class utilities $u_{1:C}$; sample temperature $\eta_s > 0$; ring buffer $\mathcal{R}$ of past divergences (size $\leq W - 1$)

**Ensure:** Replay indices $I$ with exact mass $m$ and enforced per-batch/window stealth

1: $m \leftarrow \lfloor f \cdot n_{\text{aux}} \rfloor$
2: **if** $W > 1$ **then**

3: $\quad \delta' \leftarrow \max\left\{0, \min_{0 \leq L \leq \min(W-1, |\mathcal{R}|)} \left((L+1)\delta - \sum_{\ell=|\mathcal{R}|-L+1}^{|\mathcal{R}|} \mathcal{R}[\ell]\right)\right\}$ $\qquad \triangleright$ empty sum $= 0$ for $L = 0$

4: **else**
5: $\quad \delta' \leftarrow \delta$
6: **end if**
7: **if** $\text{Div} = \text{KL}$ **then**
8: $\quad p^\star \leftarrow \textsc{ProjectKL}(p_0, u, \delta')$ $\qquad\qquad\qquad\qquad\qquad\qquad \triangleright$ Algorithm 2 (ProjectKL)
9: **else** $\qquad\qquad\qquad\qquad\qquad\qquad\qquad\qquad\qquad\qquad\qquad\qquad\qquad\qquad \triangleright \text{Div} = \text{TV}$
10: $\quad p^\star \leftarrow \textsc{ProjectTV}(p_0, u, \delta')$ $\qquad\qquad\qquad\qquad\qquad\qquad \triangleright$ Algorithm 2 (ProjectTV)
11: **end if**
12: $q \leftarrow \textsc{RoundToSum}(m\, p^\star)$
13: $q \leftarrow \textsc{ClipToAvailability}(q, \mathcal{A})$
14: $q \leftarrow \textsc{AuditAndFixQuotas}(q, p_0, \delta', \text{Div})$ $\qquad\qquad \triangleright$ unit transfers until $\text{Div}(q/m\|p_0) \leq \delta'$
15: $\bar{p} \leftarrow q/m$
16: $I \leftarrow \emptyset$
17: **for** each class $c$ **do**
18: $\quad$ add $q[c]$ indices of class $c$ by top-$\tilde{u}_i$ or by sampling $\propto \exp(\eta_s \tilde{u}_i)$ without replacement
19: **end for**
20: push $\text{Div}(\bar{p}\|p_0)$ to $\mathcal{R}$
21: **if** $|\mathcal{R}| > W - 1$ **then**
22: $\quad$ pop oldest
23: **end if**
24: **return** $I$

---

**Step A: Preference (tilt).** Encode harm preference using per-class utilities $u_c$ and a tilt strength $\eta > 0$. Define the (unconstrained) tilted mix $\tilde{p}_c \propto p_{0,c}\, e^{\eta u_c}$. For **TV** divergence, only the *ordering* of $u_c$ matters in the next step; the **KL** solution is determined exactly by the projector (Step B). When $u$ is constant, tilting has no effect and $\tilde{p} = p_0$.

**Step B: Projection to the stealth ball (exact optimizers).** We find the optimal proportions $p^\star$ that solve Eq. 2, attaining the maximum harm while satisfying $\text{Div}(p^\star\|p_0) \leq \delta'$. The method differs for KL and TV:

- **KL ball (single-tilt search).** By Lagrangian optimality, the optimizer has exponential form

$$p^\star(c) \;=\; \frac{p_0(c)\, \exp(\alpha\, u_c)}{\sum_j p_0(j)\, \exp(\alpha\, u_j)}, \tag{3}$$

  where a scalar $\alpha \geq 0$ is chosen so that $\text{KL}(p^\star\|p_0) = \delta'$ whenever the constraint is active; $\text{KL}(p_\alpha\|p_0)$ is monotone in $\alpha$ when $u$ is non-constant, so a safe bisection finds $\alpha$ (Algorithm 2 (ProjectKL)). Each evaluation is $O(C)$; overall cost is $O(C \cdot \text{iters})$.

- **TV ball (two-sided water-filling).** Sort classes by $u_c$ and greedily move probability mass from the *lowest*-utility classes to the *highest*-utility classes until the total $\ell_1$ budget $\sum_c |p_c - p_{0,c}| = 2\delta'$ is exhausted, respecting simplex bounds (Algorithm 2 (ProjectTV)). This is an exact solution and is inherently "greedy constraint" in nature: it spends the limited TV budget only on transfers that maximally increase $u^\top p$.

---

**Algorithm 2** Projection Solvers for KL and TV Balls

| **ProjectKL (left panel):** $\text{PROJECTKL}(p_0, u, \delta')$ | **ProjectTV (right panel):** $\text{PROJECTTV}(p_0, u, \delta')$ |
|---|---|
| 1: **function** $\text{TILT}(\alpha)$: | 1: $p \leftarrow p_0$; $b \leftarrow 2\delta'$ |
| 2:    $p_\alpha \leftarrow p_0 \exp(\alpha u)$; **return** $p_\alpha / \sum p_\alpha$ | 2: Sort indices: $u_{(1)} \leq \cdots \leq u_{(C)}$ |
| 3: **if** $\max u = \min u$ **then return** $p_0$ **end if** | 3: $\ell \leftarrow 1$ (donor), $r \leftarrow C$ (receiver) |
| 4: $\alpha_{\text{lo}} \leftarrow 0$; $\alpha_{\text{hi}} \leftarrow 1$ | 4: **while** $b > 0$ and $\ell < r$ **do** |
| 5: **while** $\text{KL}(\text{TILT}(\alpha_{\text{hi}})\|p_0) < \delta'$ **do** | 5:    $\varepsilon \leftarrow \min\{p_{(\ell)}, 1 - p_{(r)}, b/2\}$ |
| 6:    $\alpha_{\text{hi}} \leftarrow 2\,\alpha_{\text{hi}}$ | 6:    $p_{(\ell)} \leftarrow p_{(\ell)} - \varepsilon$ |
| 7: **end while** | 7:    $p_{(r)} \leftarrow p_{(r)} + \varepsilon$ |
| 8: **while** $\alpha_{\text{hi}} - \alpha_{\text{lo}} > \varepsilon_\alpha$ **do** | 8:    $b \leftarrow b - 2\varepsilon$ |
| 9:    $\alpha \leftarrow (\alpha_{\text{lo}} + \alpha_{\text{hi}})/2$; $p_\alpha \leftarrow \text{TILT}(\alpha)$ | 9:    **if** $p_{(\ell)} = 0$ **then** $\ell \leftarrow \ell + 1$ |
| 10:    **if** $\text{KL}(p_\alpha\|p_0) < \delta'$ **then** | 10:    **end if** |
| 11:      $\alpha_{\text{lo}} \leftarrow \alpha$ | 11:    **if** $p_{(r)} = 1$ **then** $r \leftarrow r - 1$ |
| 12:    **else** | 12:    **end if** |
| 13:      $\alpha_{\text{hi}} \leftarrow \alpha$ | 13: **end while** |
| 14:    **end if** | 14: **return** $p^\star \leftarrow p$ |
| 15: **end while** | 15: |
| 16: **return** $p^\star \leftarrow \text{TILT}(\alpha_{\text{hi}})$ | 16: |

---

**Both methods return the exact optimizer $p^\star$ of Eq. 2.**

**Step C: Budgeted quotas and within-class selection.**

Convert $p^\star$ to practical integer quotas $q \in \mathbb{N}_0^{C_t}$ (Alg. 1, Lines 9–11): (i) apply *largest-remainder rounding* (Hamilton method (Janson & Linusson, 2012)): set $q_c \leftarrow \lfloor m\, p_c^\star \rfloor$, then allocate the remaining $m - \sum_c q_c$ units to classes with the largest fractional parts of $m\, p_c^\star$ until $\sum_c q_c = m$; (ii) *clip to availability* ($q_c \leq a_c$) and redistribute any deficit while preserving the total mass $m$; (iii) run *audit-and-fix* to ensure $\text{Div}(q/m\|p_0) \leq \delta'$.

Rounding introduces only tiny, explicitly bounded distortions: $\|\bar{p} - p^\star\|_\infty \leq 1/m$ and $\|\bar{p} - p^\star\|_1 \leq C_t/m$ (i.e., the bound scales with the current number of seen classes).

For **TV**, each single-unit transfer from any class with $\bar{p}_i > p_{0,i}$ to any class with $\bar{p}_j < p_{0,j}$ decreases TV by exactly $1/m$, so feasibility is reached in finitely many swaps (at most $2\delta'm$ swaps in the worst case). For **KL**, we perform discrete "steepest-decrease" swaps (from the largest log-ratio $\log(\bar{p}_c/p_{0,c})$ to the smallest), which strictly decreases KL each step until $\text{KL}(\bar{p}\|p_0) \leq \delta'$. Within each class, select indices by *top-$\tilde{u}_i$* or *probabilistically* with $\Pr(i\,|\,y_i = c) \propto e^{\eta_s \tilde{u}_i}$ (no replacement).

**Step D: Windowed visibility (online scheduler).** With windowed auditing (length $W$), we compute a tightened $\delta_t$ from the ring buffer $\mathcal{R}$ (Alg. 1, Lines 2–6) and enforce $\text{Div}(\bar{p}_t\|p_0) \leq \delta'_t$ at each step. Intuitively, the scheduler spends only the *residual* budget left after accounting for the last $W-1$ steps. In ideal arithmetic, this implies a deterministic sliding-window guarantee:

**Proposition (Residual-budget window compliance).** Let $\delta'_t$ be chosen as in Alg. 1 and enforce $\text{Div}(\bar{p}_t\|p_0) \leq \delta'_t$ at every step. If Div is convex in its first argument, then for all $t$ and all $1 \leq L \leq W$,

$$\text{Div}\big(\hat{p}_{t-L+1:t} \,\|\, p_0\big) \;\leq\; \delta. \tag{4}$$

*Proof sketch.* The residual update ensures the partial sums over any trailing window of length $L \leq W$ never exceed $L\delta$. By Jensen/convexity, $\text{Div}(\hat{p}\|p_0) \leq \frac{1}{L}\sum_s \text{Div}(\bar{p}_s\|p_0) \leq \delta$.

In practice, discretization/availability can introduce rare numerical slack; we therefore also report empirical window-violation rates in §3.4.

**Complexity.** Scheduler update is $O(W)$. Projection is $O(C \cdot \text{iters})$ (KL) or $O(C \log C)$ (TV, for sorting). Quotas/auditing are $O(C)$ plus a small number of unit transfers.

## 3.4 Visibility and Efficiency Guarantees

Our method provides three key guarantees by construction, ensuring stealth and budget compliance; we also state an explicit dominance property.

**Guarantee 1: Per-batch Stealth (hard constraint) and tail reporting (policy).** By Step C's audit-and-fix, the realized histogram satisfies $\text{Div}(\bar{p}_t \| p_0) \leq \delta'$ (hence $\leq \delta$). For reporting, we define the *normalized* per-batch divergence and its 95th-percentile summary:

$$r_t := \frac{\text{Div}(\bar{p}_t \| p_0)}{\delta} \in [0, 1], \qquad r_{\text{batch@95}} := \text{Quantile}_{0.95}\{r_t\}. \tag{5}$$

We use $r_{\text{batch@95}} \leq 0.05$ as a *calibrated reporting threshold*: it means 95% of batches spend at most 5% of the audit radius, leaving headroom for rounding/availability noise. This is a policy choice for distinguishing "highly stealthy" regimes (not a requirement implied by the definition of the audit ball). Appendix reports the clean-run calibration motivating the 0.05 band.

**Guarantee 2: Windowed Stealth (deterministic scheduler) and empirical slack.** Under Step D's residual-budget scheduler and convexity of Div, Eq. 4 holds for any window $L \leq W$. We additionally report an empirical **window violation rate** to capture any residual exceedances due to discretization/measurement noise:

$$r_{\text{win}} = \frac{1}{T} \sum_{t=1}^{T} \mathbb{1}\!\!\!\!/\left\{\text{Div}(\hat{p}_{t-W+1:t} \| p_0) > \delta\right\}. \tag{6}$$

We treat $r_{\text{win}} \leq 0.05$ as an *operational acceptance band* (clean-calibrated) rather than a theoretical necessity.

**Guarantee 3: Budget Conservation.** The quota generation process (Step C) is strictly mass-preserving, ensuring $\sum_{c=1}^{C} q_{t,c} = m$. This yields a realized keep fraction $\hat{f}_t = m/n_{\text{aux}}$ that tightly tracks the target $f$, with $|\hat{f}_t - f| \leq 1/n_{\text{aux}}$. We verify this using the **95th-percentile fraction error**:

$$e_{95} = \text{Quantile}_{0.95}(|\hat{f}_t - f|). \tag{7}$$

Because mass is exact, we have the deterministic bound $e_{95} \leq 1/n_{\text{aux}}$ (e.g., $\leq 0.002$ for $n_{\text{aux}}=500$); we use $e_{95} \leq 0.02$ only as a loose reporting range.

**Dominance property (optimality under the audited constraint).** Fix any non-constant utility vector $u$, nominal histogram $p_0$, divergence type $\text{Div} \in \{\text{KL}, \text{TV}\}$, and active budget $\delta'$. Let $p^\star$ be the solution of Eq. 2. Then for any other feasible sampler-level class mix $p \in \Delta^C$ with $\text{Div}(p\|p_0) \leq \delta'$, we have $u^\top p \leq u^\top p^\star$, with strict inequality unless $p$ is also optimal (and in particular if $\delta' > 0$ and $u$ is non-constant, the optimizer is unique for KL). Thus, given a specified harm surrogate $u$ and the same auditable constraint set, Amnesia's projection step is not a heuristic: it is the exact best-response class composition.

## 4 Experimental Setup

### 4.1 Datasets

We evaluate on four standard CL benchmarks of increasing difficulty: *Split CIFAR-10* (Zenke et al., 2017) (10 classes, 32×32 RGB; 5 tasks × 2 classes, easy), *CORe50* (Lomonaco & Maltoni, 2017) (50 classes; class-incremental; 10 tasks × 5 classes, harder), *Split CIFAR-100* (Zenke et al., 2017) (100 classes, 32×32 RGB; 10 tasks × 10 classes), and *Tiny-ImageNet* (Stanford, 2015) (ImageNet subset, 200 classes, 64×64; we use 100 classes as 5 tasks × 20 classes and reserve the remaining 100 for auxiliary out-of-stream ablations, more challenging).

### 4.2 Replay-based Continual Learning Methods

Replay CL maintains a small memory $\mathcal{M}$ and, at step $t$, optimizes on the union of the current mini-batch $B_t$ and a memory mini-batch $M_t \subset \mathcal{M}$ (typically via reservoir sampling). We attack the canonical **Experience Replay (ER)** (Rolnick et al., 2019) and three prominent extensions: **ER-ACE** (Caccia et al., 2022) (asymmetric cross-entropy restricting logits on new data to current-task classes to reduce representation drift), **DER++** (Buzzega et al., 2020) (ER with knowledge distillation by matching current logits to stored past logits for memory samples), and **SCR** (Mai et al., 2021) (supervised contrastive loss on mixed new+replay batches to learn a more unified representation).

### 4.3 Training Protocol

**Protocol.** Unless stated otherwise, we use **ResNet-18** (He et al., 2016) with **SGD** (lr = 0.03), mini-batch size 64, and buffer size 500. Images follow official splits and are normalized to $[0, 1]$; CORe50 uses the *New Classes (NC)* scenario. Each task is trained for **10 epochs** with a **fixed** task order; results are averaged over **5** seeds. Default replay/audit settings are keep fraction $f=0.1$, stealth radius $\delta=0.1$, and audit window $W=10$.

**Clean vs. attacked hyperparameters.** For every method/dataset, learner-side hyperparameters are *identical* between clean and attacked runs (optimizer, learning rate, epochs, augmentations, buffer size, and method-specific settings), isolating the effect of *sampler-level replay composition* (index selection) from retuning. Since CL performance can be hyperparameter-sensitive, we focus on *relative* degradation under a fixed, standard protocol rather than globally optimal tuning.

**Sampler telemetry used to construct utilities.** Amnesia consumes lightweight, *sampler-readable* utility logs produced by the training job and read asynchronously by the sampler. Unless stated otherwise, $\tilde{u}_i$ is the cross-entropy loss of a replayed example when it is last observed during training, and $u_c$ is an EMA over an aggregation (mean) of $\tilde{u}_i$ for samples with label $c$; thus the sampler requires no real-time per-index model queries and only reads the latest available utility snapshot.

**Nominal histogram for auditing.** Unless stated otherwise, $p_0$ is the replay-buffer class histogram at sampling time (with standard smoothing to ensure $p_{0,c} > 0$ for KL), and all audit metrics (TV or KL) are computed between the realized replay histogram and this $p_0$.

### 4.4 Evaluation Criteria

Beyond the stealth/budget metrics in §3.4, we report final accuracy and backward transfer (BWT). Let $R_{i,j}$ denote performance on task $j$ after training through task $i$, and let $T$ be the number of tasks:

$$\text{BWT} = \frac{1}{T-1} \sum_{j=1}^{T-1} \left( R_{T,j} - R_{j,j} \right), \qquad \text{ACC} = \frac{1}{T} \sum_{j=1}^{T} R_{T,j}. \tag{8}$$

By convention, negative BWT indicates forgetting; we therefore report $-$BWT so that larger values correspond to more forgetting (stronger attack impact).

## 5 Results

**Main impact and audit adherence.** Table 2 summarizes *impact* (ACC↓, $-$BWT ↑) and *audit adherence* (Stealth, Budget). The sampler enforces the core per-batch constraint $\text{Div}(\bar{p}\|p_0) \le \delta'$ by construction (via audit-and-fix); red denotes exceeding conservative reporting bands: (i) $r_{\text{batch@95}} \le 0.05$ (95% of steps spend at most 5% of the audit radius; tail headroom) and (ii) $r_{\text{win}} \le 0.05$ (an operational target for rare window exceedances under discretization/availability noise). Under these criteria, green highlights the best *compliant* entries per setting.

Across datasets and replay methods, Amnesia induces substantial forgetting: ACC drops sharply and $-$BWT rises. Canonical ER is often (though not always) among the most vulnerable. ER-ACE, which mitigates representation drift, is typically the most robust *among compliant runs* (e.g., Tiny-ImageNet/KL) while still suffering large degradation. DER++ is particularly sensitive in high-class regimes (CIFAR-100, Tiny-ImageNet), exhibiting both strong impact and frequent red-flagged stealth/budget issues. A plausible explanation is that DER++'s distillation loss sharpens the utility landscape and creates more extreme sampling pressures, which are harder to realize cleanly under integer quotas and limited per-class availability.

**When and why do red-flagged audit metrics occur?** Band exceedances are more common as the *effective granularity* tightens, especially when $m/C \approx 1$ (few items per class per batch). CORe50 ($C=50$) is

Table 2: **Amnesia attack results (Impact / Stealth / Budget).** Base (no attack) vs. **KL** and **TV** for all models/datasets. **Impact:** ACC↓, −BWT ↑; **Stealth:** $r_{\text{batch@95}}$ ↓, $r_{\text{win}}$ ↓; **Budget:** $e_{95}$ ↓. *Mean±std over 5 seeds. Stealth/Budget are ×10$^{-2}$ (e.g., 5=0.05). Red cells exceed conservative, clean-calibrated reporting bands ($r_{batch@95} \leq 0.05$, $r_{win} \leq 0.05$, $e_{95} \leq 0.02$); these are reporting policies (tail headroom / violation-rate targets), not a change to the enforced per-batch audit constraint* $\text{Div}(\bar{p}\|p_0) \leq \delta'$ *(guaranteed by construction).*

| Dataset | Model | No Attack ACC↓ (-BWT↑) | KL Impact ACC↓ (-BWT↑) | KL Stealth $r_{\text{batch@95}}$ ↓ ($r_{\text{win}}$ ↓) | KL Budget $e_{95}$ ↓ | TV Impact ACC↓ (-BWT↑) | TV Stealth $r_{\text{batch@95}}$ ↓ ($r_{\text{win}}$ ↓) | TV Budget $e_{95}$ ↓ |
|---|---|---|---|---|---|---|---|---|
| CIFAR-10 | ER | 50.4±0.6 (55.1±1.3) | 29.3±0.29 (82.7±2.0) | 5.0±0.4 (0.2±0.1) | 1.0±0.3 | 25.1±0.25 (88.2±2.3) | 5.0±2.0 (6.0±1.5) | 1.5±0.5 |
| | **SCR** | 59.6±0.7 (38.9±0.4) | **31.1±0.8 (78.2±2.1)** | **4.0±0.3 (0.7±0.2)** | **1.5±0.4** | 28.9±0.29 (83.4±2.3) | 4.0±0.3 (0.7±0.2) | 1.8±0.5 |
| | DER++ | 64.0±0.7 (29.1±0.3) | 33.1±0.9 (69.1±2.0) | 5.0±0.5 (0.2±0.1) | 0.9±0.3 | 30.7±0.90 (75.1±2.2) | 10±4.0 (1.0±0.5) | 2.0±0.7 |
| | ER-ACE | 65.2±0.8 (15.2±0.15) | 34.2±1.0 (56.9±1.7) | 1.0±0.2 (0.1±0.1) | 1.0±0.3 | 32.3±0.95 (60.3±1.8) | 4.0±0.4 (3.0±0.8) | 2.0±0.7 |
| CORe50 | **ER** | 55.4±0.7 (52.5±1.5) | 32.5±0.8 (84.4±2.1) | 0.8±0.2 (0±0.0) | 0.9±0.3 | **32.1±0.96 (85.8±2.2)** | **4.0±0.3 (0±0.0)** | **0.8±0.3** |
| | SCR | 52.9±0.8 (58.8±1.6) | 28.5±0.28 (87.2±2.2) | 5.0±0.4 (0±0.0) | 0.9±0.3 | 33.5±1.00 (82.4±2.1) | 3.0±0.3 (5.0±0.8) | 0.9±0.3 |
| | DER++ | 61.7±0.8 (47.3±0.47) | 30.1±0.90 (86.8±2.2) | 8.0±3.0 (0.1±0.5) | 1.0±0.6 | 32.2±0.96 (84.53±2.4) | 10±4.0 (0.3±1.0) | 1.0±0.8 |
| | ER-ACE | 68.7±0.9 (24.5±0.25) | 36.0±1.0 (75.7±2.0) | 1.4±0.4 (0.1±0.1) | 2.0±0.8 | 34.6±1.00 (74.7±2.1) | 4.0±1.5 (3.0±2.0) | 0.9±0.6 |
| CIFAR-100 | ER | 35.2±0.7 (57.4±1.7) | 20.3±0.20 (72.7±2.2) | 4.0±0.6 (2.0±0.6) | 0.3±0.2 | 21.8±0.22 (73.1±2.2) | 5.0±0.7 (3.0±0.6) | 1.0±0.5 |
| | SCR | 33.5±0.7 (52.3±1.6) | 21.5±0.22 (75.6±2.3) | 4.0±0.5 (3.0±0.6) | 1.2±0.7 | 19.2±0.19 (81.4±2.4) | 15±4.0 (2.0±1.0) | 1.9±0.9 |
| | DER++ | 33.1±0.7 (45.2±0.45) | 18.7±0.19 (63.6±1.9) | 13±5.0 (2.0±1.0) | 1.9±0.9 | 20.1±0.20 (60.5±1.8) | 41±5.0 (6±5.0) | 5.0±3.0 |
| | **ER-ACE** | 44.2±0.9 (21.3±0.21) | 23.3±0.23 (65.6±2.0) | 3.7±0.6 (0±0.0) | 0.2±0.2 | **21.5±0.22 (70.1±2.1)** | **5.0±0.9 (5.0±0.9)** | **0.9±0.8** |
| TinyImageNet | ER | 17.0±0.17 (21.3±0.21) | 8.2±0.08 (39.3±0.39) | 4.0±0.6 (3.0±0.6) | 0.6±0.5 | 6.5±0.06 (45.9±0.45) | 11±4.0 (12±5.0) | 1.2±0.8 |
| | SCR | 22.5±0.22 (20.2±0.20) | 11.4±0.11 (33.7±0.33) | 12±4.0 (2.0±1.0) | 1.1±0.8 | 9.9±0.09 (42.8±0.42) | 5.0±0.8 (4.0±0.8) | 2.0±0.9 |
| | DER++ | 21.8±0.22 (19.5±0.20) | 10.2±0.10 (30.1±0.30) | 11±3.0 (2.0±1.0) | 1.2±0.8 | 11.8±0.12 (38.4±0.38) | 34±5.0 (36±5.0) | 5.1±3.0 |
| | **ER-ACE** | 23.5±0.24 (15.7±0.16) | **13.1±0.13 (45.1±0.45)** | **2.8±0.6 (1.0±0.6)** | **0.7±0.5** | 10.5±0.10 (52.4±1.50) | 5.0±0.7 (4.0±0.8) | 1.9±0.8 |

representative when $f=0.1$ (small $m$), and CIFAR-100 / Tiny-ImageNet are also tight because $C$ is large. In these regimes, discretization (rounding), clipping to availability, and sparse allocations (especially for TV) can create batch-to-batch spikes in normalized divergence and rare window exceedances. This matches §3.3: when $m/C$ is small, even a one-unit quota change yields a large change in $\bar{p}$, so realized histograms become inherently "chunky," increasing the risk of tail spikes in $r_t$ and, for TV, occasional window exceedances under aggressive reallocation.

**KL vs. TV trade-off and the "greedy constraint" question.** A consistent KL–TV trade-off emerges: TV often yields slightly stronger impact but triggers more red-flagged stealth/budget entries. This follows from the projectors. KL's exponential tilt, $p^\star(c) \propto p_0(c) \exp(\alpha u_c)$, preserves positive mass on all classes and discretizes more smoothly, producing fewer tail/window spikes. TV's two-sided water-filling can push low-utility classes toward 0, producing a sparser and more extreme $p^\star$ (closer to the unconstrained optimum) that is harder to realize under integerization and availability constraints.

This also motivates a simple, budget-aware baseline: *the TV variant is itself greedy and budget-aware.* The TV projector is exactly two-sided water-filling, greedily transferring probability mass from the lowest-utility to the highest-utility classes until the TV budget is exhausted (Algorithm 2 (ProjectTV)). For KL, the "optimization" is a single-parameter monotone search implementing the exponential-tilt solution (Algorithm 2 (ProjectKL)). Thus, the key modeling choice is the harm surrogate $u$ (defined in §3.1 and instantiated in §4.1).

## 5.1 Robustness to Backbone Choice

Replacing ResNet-18 with ViT-Tiny and ConvNeXt-Base (CIFAR-100, ER-ACE, KL) preserves the qualitative conclusion (Table 3): Amnesia remains strongly effective (large ACC drops and severe increases in forgetting) while staying within the same audit policy (low $r_{\text{batch@95}}$, zero $r_{\text{win}}$, and tight $e_{95}$). Stronger backbones improve clean performance and show slightly higher robustness under attack (higher attacked ACC and slightly lower forgetting), suggesting that increased representational stability can reduce (but not eliminate) the leverage of divergence-constrained replay steering.

**Ablation insights (ER-ACE, Split CIFAR-10).** Table 4 disentangles **Preference** (aux_trim) and **Projection** (PrO). *Preference without Projection* (**PO**) can improve clean CL (higher ACC, lower −BWT):

Table 3: **CIFAR-100 (ER-ACE) under (KL attack)**. Values are mean±std over 5 seeds. Stealth and Budget are reported $\times 10^{-2}$ (e.g., 5 denotes 0.05).

| Metric | VIT | Convnext |
|---|---|---|
| Baseline ACC | $48.1_{\pm 0.8}$ | $51.7_{\pm 0.9}$ |
| Baseline Forget | $20.0_{\pm 0.20}$ | $19.1_{\pm 0.18}$ |
| Attack ACC | $25.6_{\pm 0.25}$ | $27.9_{\pm 0.30}$ |
| Attack Forget | $63.8_{\pm 1.8}$ | $61.9_{\pm 1.7}$ |
| Stealthy $r_{\text{batch@95}}$ | $3.5_{\pm 0.6}$ | $3.3_{\pm 0.5}$ |
| Stealthy $r_{\text{win}}$ | $0_{\pm 0.0}$ | $0_{\pm 0.0}$ |
| Budget $e_{95}$ | $0.2_{\pm 0.2}$ | $0.2_{\pm 0.2}$ |

Table 4: **Ablation of preference and projection on ER-ACE (Split CIFAR-10).** ✓/✗ denote enabled/disabled. **aux_trim** utility prioritization; **PrO** is the projection only **PO**: aux_trim only.

| Model | aux_trim | PrO | Div. | ACC (-BWT) | $r_{\text{batch@95}}(r_{\text{win}})$ | $e_{95}$ |
|---|---|---|---|---|---|---|
| ER-ACE (Baseline) | ✗ | ✗ | N/A | 65.21 / 15.25 | N/A | N/A |
| PO | ✓ | ✗ | N/A | 68.55 / 12.39 | N/A | N/A |
| PrO (KL) | ✗ | ✓ | KL | 45.07 / 31.67 | 0.516 (0.224) | 0.013 |
| PrO (TV) | ✗ | ✓ | TV | 47.38 / 26.06 | 1.183 (0.222) | 0.012 |
| Amnesia (TV) | ✓ | ✓ | TV | 32.34 / 60.30 | 0.040 (0.085) | 0.020 |
| **Amnesia (KL)** | ✓ | ✓ | **KL** | **34.20 / 56.90** | **0.010 (0.000)** | **0.011** |

PO resembles prioritized replay / hard-example mining and does not enforce an adversarial class-level reallocation under an audit constraint, so it can act as a benign optimization rather than an attack. *Projection without Preference* (**PrO**) increases forgetting but is substantially more visible (large $r_{\text{batch@95}}$ and nontrivial $r_{\text{win}}$), showing that spending divergence budget without harm-targeted preference is inefficient and detectable. The full two-stage method restores the intended harm–stealth balance: **Amnesia (KL)** achieves strong forgetting with clean batch/window compliance, while **Amnesia (TV)** is more aggressive but generally more brittle because sparse, extreme allocations are harder to realize under discretization and availability constraints.

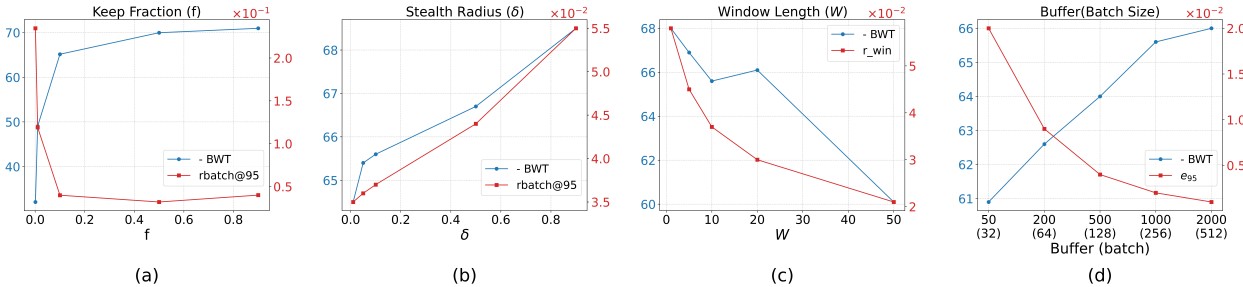

(a)  (b)  (c)  (d)

Figure 2: **Ablations (ER-ACE, Split CIFAR-10).** Blue (left axis): $-$BWT (impact). Red (right axis): an audit/budget metric. (a) $x$: keep fraction $f$; red: $r_{\text{batch@95}}$ ($\times 10^{-1}$). (b) $x$: stealth radius $\delta$; red: $r_{\text{batch@95}}$ ($\times 10^{-2}$). (c) $x$: audit window $W$; red: $r_{\text{win}}$ ($\times 10^{-2}$). (d) $x$: buffer size (batch size in parentheses); red: $e_{95}$ ($\times 10^{-2}$).

Fig. 2(a,b) show keep fraction $f$ and stealth radius $\delta$ effects (blue: higher $-$BWT $\Rightarrow$ more forgetting; red: batch-level audit pressure). In Fig. 2(a), increasing $f$ enlarges replay mass $m = \lfloor fn_{\text{aux}} \rfloor$, letting the sampler realize $p^\star$ more faithfully and select more harmful items; thus $-$BWT rises, while $r_{\text{batch@95}}$ falls due to reduced discretization pressure, matching Step C's bound $\|\bar{p} - p^\star\|_1 \le C/m$ (larger $m \Rightarrow$ less "chunkiness"). In Fig. 2(b), increasing $\delta$ expands the feasible ball, so the optimizer spends more divergence and both $-$BWT and $r_{\text{batch@95}}$ increase monotonically (harm–stealth trade-off).

Fig. 2(c) shows larger $W$ makes the residual-budget scheduler more conservative (smaller $\delta_t$), slightly reducing $-$BWT while driving window violation rate down, consistent with the window-compliance guarantee in §3.4. Fig. 2(d) shows larger buffers strengthen the attack (more replay candidates and stronger within-class choice) while improving budget tracking, since larger buffers/batches reduce integerization effects and availability-driven failure modes.

Because prior baselines do not enforce auditable stealth/budget constraints, Table 5 reports only shared metrics (ACC, $-$BWT) and should be read as *impact-at-any-cost* references. In task-incremental CIFAR-10 with ER, we compare Amnesia with three data-poisoning baselines; Table 5 lists their ACC and $-$BWT. Despite operating under explicit audit-aware constraints and sampler-only access (pixels/labels/parameters

Table 5: Comparison between Amnesia and similar attacks.

| Attack Method | Result |
|---|---|
| Targeted Poisoning (Li & Ditzler, 2022) | 19.6 (66.3) |
| PACOL (Umer et al., 2023) | 15.8 (68.7) |
| BrainWash (Abbasi et al., 2024) | 25.5 (75.4) |
| **Ours** | **29.3 (82.7)** |

Table 6: End-to-end training time on CIFAR-10 (mean±std).

| Method | Time (min) |
|---|---|
| Baseline (no attack) | 11:34 ± 0:28 |
| Amnesia (KL) | 11:58 ± 0:49 |
| Amnesia (TV) | 12:23 ± 0:58 |

untouched), Amnesia achieves strong forgetting, highlighting index-only replay control as a practical threat surface in monitored deployments.

**Runtime overhead.** Relative to baseline, Amnesia adds modest end-to-end cost: $+24\,$s ($\approx 3.5\%$) for KL and $+49\,$s ($\approx 7.1\%$) for TV. This matches sampler-side costs: $O(W)$ scheduling, $O(C)$ quotas/audit, and $O(C)$ (KL) vs. $O(C \log C)$ (TV) projection, which remain small relative to forward/backward passes and support the practicality of sampler-only interference.

Appendix B.3 (Table 10) provides a component-level breakdown of this overhead into (i) projection, (ii) Rounding/Clipping, and (iii) Audit-and-Fix, highlighting that TV is dominated by sorting in projection and by swap counts in audit-and-fix.

## 6 Conclusion

We identified a realistic, auditable vulnerability in continual learning: *sampler-level* control of replay indices. **Amnesia** casts malicious replay composition as a divergence-constrained program with explicit visibility ($\delta$) and mass ($f$) budgets, combining a harm-driven *preference* step with an exact *projection* onto a TV/KL stealth ball. The resulting optimizers, a KL *single-tilt* and a TV *two-sided water-filling*, together with a windowed scheduler, are efficient, mass-preserving, and optimal within the audited set. Across Split CIFAR-10/100, CORe50, and TinyImageNet, with strong replay baselines (ER, ER-ACE, SCR, DER++), Amnesia reliably reduces accuracy and increases forgetting while satisfying audits in most settings. KL yields near-maximal damage with high compliance; TV achieves higher impact but is brittle when the mass-per-class ratio is tight. Ablations confirm that both *preference* and *projection* are necessary to attain the intended impact–visibility–budget trade-off.

A limitation of this study is that it assumes labeled buffers and a fixed nominal histogram $p_0$ for auditing; applicability to unlabeled or evolving label spaces remains unexplored. Moreover, when $m/C \approx 1$, discretization and availability constraints can strain stealth, especially for TV. Future directions can develop sampler-aware defenses (attested index selection, cryptographic logging, randomness beacons) and *multi-metric* auditors beyond class histograms (e.g., MMD/CUSUM, gradient telemetry), and extend the framework to generative or unlabeled replay, task-free online CL, and RL/robotics, alongside theory linking $(\delta, f)$ to bounds on expected $-$BWT and detectability. More broadly, future work should test whether analogous selection/composition attack surfaces arise in non-replay CL and continual learning for language models (e.g., retrieval/memory selection, prompt/example selection, or routing). Our findings elevate replay index selection to a first-class security primitive, underscoring the need to secure the data path, not just the model, in deployed CL systems.

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

# A    Methodology Proofs

## A.1    KL projector: optimality and monotone root search

**Lemma 1** (KL-ball optimizer). *For $p_0 \in \Delta^C$ with $p_{0,c} > 0$, $u \in \mathbb{R}^C$, and $\delta' \geq 0$, the problem*

$$\max_{p \in \Delta^C} u^\top p \ s.t. \ \mathrm{KL}(p\|p_0) \leq \delta'$$

*admits an optimizer of the form*

$$p^\star(c) = \frac{p_{0,c} e^{\alpha u_c}}{\sum_j p_{0,j} e^{\alpha u_j}}$$

*for some $\alpha \geq 0$. If $u$ is non-constant, this optimizer is unique. If $u$ is constant, any feasible $p$ is optimal and the choice $\alpha = 0$ yields $p^\star = p_0$.*

*Sketch.* Lagrangian $\mathcal{L}(p, \lambda, \nu) = u^\top p - \lambda(\sum_c p_c - 1) - \nu(\mathrm{KL}(p\|p_0) - \delta')$, $\nu \geq 0$. Stationarity: $u_c - \lambda - \nu(\log \frac{p_c}{p_{0,c}} + 1) = 0 \Rightarrow \log \frac{p_c}{p_{0,c}} = \alpha u_c + \beta$, so $p_c \propto p_{0,c} e^{\alpha u_c}$. Strict convexity of $\mathrm{KL}(\cdot\|p_0)$ on the simplex implies uniqueness when $u$ is non-constant; if $u$ is constant, $u^\top p$ is constant on the feasible set and any feasible $p$ is optimal (the tilt with $\alpha = 0$ returns $p_0$). $\square$

**Lemma 2** (Monotonicity of the KL radius). *Let $p_\alpha(c) = \dfrac{p_{0,c} e^{\alpha u_c}}{\sum_j p_{0,j} e^{\alpha u_j}}$ and $g(\alpha) = \mathrm{KL}(p_\alpha\|p_0)$. Then*

$$g'(\alpha) \ = \ \alpha \, \mathrm{Var}_{p_\alpha}(u) \ \geq \ 0,$$

*with $>$ for $\alpha > 0$ when $u$ is non-constant. Thus $g$ is strictly increasing on $(0, \infty)$ and bisection/Newton finds the unique $\alpha$ with $g(\alpha) = \delta'$.*

*Sketch.* $g(\alpha) = \alpha \mathbb{E}_{p_\alpha}[u] - \log Z(\alpha)$, where $Z(\alpha) = \sum_c p_{0,c} e^{\alpha u_c}$. Using $\frac{d}{d\alpha} \log Z = \mathbb{E}_{p_\alpha}[u]$ and $\frac{d^2}{d\alpha^2} \log Z = \mathrm{Var}_{p_\alpha}(u)$, we get $g'(\alpha) = \alpha \mathrm{Var}_{p_\alpha}(u)$. $\square$

The KL-constrained problem says: maximize a linear score $u^\top p$ while staying close to $p_0$ in KL. The Lagrange multiplier conditions force the optimal solution to have *log-ratios* $\log \frac{p_c}{p_{0,c}}$ that are linear in utilities $u_c$, which is exactly the *exponential tilt* $p_c \propto p_{0,c} e^{\alpha u_c}$. Because KL is strictly convex, this tilted form is the unique optimizer on the KL ball (when $u$ is non-constant). As $\alpha$ increases, the solution places progressively more mass on higher-utility classes; one can show the KL radius grows like $\alpha \times \mathrm{variance}(u)$ under the tilted distribution, which is strictly increasing unless $u$ is constant. Hence a simple one-dimensional search (bisection/Newton) finds the unique $\alpha$ that hits the budget.

## A.2    TV projector: optimality of two-sided water-filling

With a linear objective and an $\ell_1$ distance budget to $p_0$, the best way to improve $u^\top p$ is always to move probability from *worse* classes to *better* classes. Any plan that moves mass from a not-so-bad donor to a not-so-good receiver can be improved by instead using the *lowest* utility donor and the *highest* utility receiver for the same budget cost. This "exchange" logic justifies sorting once by $u_c$ and then repeatedly transferring mass from the bottom to the top until the budget is exhausted or you hit the $[0, 1]$ bounds—exactly the water-filling procedure.

**Lemma 3** (TV-ball optimizer). *For $\delta' \geq 0$, the solution to $\max_{p \in \Delta^C} u^\top p$ subject to $\frac{1}{2}\|p - p_0\|_1 \leq \delta'$ is obtained by moving probability from the lowest-u classes to the highest-u classes until the $\ell_1$ budget $2\delta'$ is exhausted or box constraints hit.*

*Sketch.* This is a linear program over a simplex slice. If any feasible $p$ moves mass from a donor $i$ to a receiver $j$ with $u_i > u_{i'}$ or $u_j < u_{j'}$, swapping to use the *lowest* donor and *highest* receiver improves $u^\top p$ without changing feasibility (standard exchange argument). Sorting once and greedily transferring realizes the optimum. $\square$

### A.3 Rounding error bounds (largest remainder / Hamilton)

We turn real-valued target counts $mp^\star$ into integers by taking floors and then giving the leftover items to the classes with the largest fractional parts. This guarantees each class's realized fraction $\bar{p}_c = q_c/m$ is at most one item off from its target, i.e., $|\bar{p}_c - p_c^\star| \leq 1/m$. Since at most $C$ coordinates can differ and each by at most $1/m$, the total $\ell_1$ error is at most $C/m$. In short, rounding introduces only *tiny*, explicitly bounded distortions.

**Lemma 4** (Rounding bounds). *Let $p^\star \in \Delta^C$, $m \in \mathbb{N}$, $q_c = \lfloor mp_c^\star \rfloor$, and assign the $m - \sum_c q_c$ leftover units to the largest fractional remainders of $mp_c^\star$. With $\bar{p} = q/m$,*

$$\|\bar{p} - p^\star\|_\infty \leq \frac{1}{m}, \qquad \|\bar{p} - p^\star\|_1 \leq \frac{C}{m}.$$

*Proof.* Each coordinate changes by at most $1/m$; at most $C$ coordinates change by $1/m$. Summing yields the $\ell_1$ bound; the $\ell_\infty$ bound is immediate. $\square$

### A.4 Audit-and-Fix: termination and decrease

---
**Algorithm 3** AUDITFIXTV$(q, p_0, \delta')$
---
1: $\bar{p} \leftarrow q/m$
2: **while** $\mathrm{TV}(\bar{p}\|p_0) > \delta'$ **do**
3:      pick donor $i$ with $\bar{p}_i > p_{0,i}$ and receiver $j$ with $\bar{p}_j < p_{0,j}$
4:      $q_i \leftarrow q_i - 1$; $q_j \leftarrow q_j + 1$; $\bar{p} \leftarrow q/m$
5: **end while**
6: **return** $q$
---

**TV case.** *TV:* If your realized mix $\bar{p}$ strays past the TV budget, moving *one* item from any class that is *above* its nominal level $p_{0,c}$ to any class that is *below* nominal shrinks the $\ell_1$ gap by exactly $2/m$, so TV drops by $1/m$ every move. After finitely many item swaps, you must re-enter the budget. *KL:* KL is a sum of convex terms $x \log(x/p_0)$. Moving one item from the class with the largest log-ratio $\log(\bar{p}_c/p_{0,c})$ to the class with the smallest log-ratio makes the KL strictly smaller. Repeating this discrete "steepest decrease" step guarantees KL falls below the threshold in finitely many swaps.

**Lemma 5** (Decrease & termination for TV). *Each unit transfer in Alg. 3 decreases $\|\bar{p} - p_0\|_1$ by $2/m$, hence TV by $1/m$. Therefore the loop terminates in at most*

$$N_{\max} = \left\lceil \frac{(\|\bar{p}^{(0)} - p_0\|_1 - 2\delta')_+}{2/m} \right\rceil$$

*transfers.*

*Proof.* Moving $1/m$ from a coordinate above $p_{0,i}$ to one below $p_{0,j}$ reduces the two absolute deviations by $1/m$ each; others unchanged. $\square$

---
**Algorithm 4** AUDITFIXKL$(q, p_0, \delta')$
---
1: $\bar{p} \leftarrow q/m$
2: **while** $\mathrm{KL}(\bar{p}\|p_0) > \delta'$ **do**
3:      donor $i \in \arg\max_c \log\frac{\bar{p}_c}{p_{0,c}}$;    receiver $j \in \arg\min_c \log\frac{\bar{p}_c}{p_{0,c}}$
4:      $q_i \leftarrow q_i - 1$; $q_j \leftarrow q_j + 1$; $\bar{p} \leftarrow q/m$
5: **end while**
6: **return** $q$
---

**KL case.**

**Lemma 6** (Strict decrease for KL (discrete step)). *Let $s = 1/m$. If $\bar{p}_i > p_{0,i}$ and $\bar{p}_j < p_{0,j}$, then for $p' = \bar{p} - se_i + se_j$,*

$$\text{KL}(p'\|p_0) - \text{KL}(\bar{p}\|p_0) = h(p_i - s) - h(p_i) + h(p_j + s) - h(p_j) < 0,$$

*where $h(x) = x\log(x/p_0)$ is convex (coordinatewise). Choosing $i, j$ by extreme log-ratios ensures decrease until feasibility $\text{KL} \leq \delta'$ holds, so Alg. 4 terminates in finitely many steps.*

*Sketch.* Convexity of $h$ implies the discrete move from an over-weighted to an under-weighted coordinate reduces the sum $\sum_c h(p_c)$; picking extremes gives the steepest decrease among unit moves. □

### A.5 Windowed scheduler guarantee

Divergence is convex, so the divergence of a window *average* of batches is at most the *average* of their divergences. The scheduler keeps a running ledger of how much divergence has already been "spent" in recent steps and assigns the next step only the *residual* budget so that, for each window length $L \leq W$, the cumulative divergence over the last $L-1$ steps plus the current step is bounded by $L\delta$ (conservative), which implies the window-average divergence stays within $\delta$.

**Lemma 7** (Residual-budget scheduler implies window stealth). *Let $r_t = \text{Div}(\bar{p}_t\|p_0)$ and define*

$$\delta'_t = \max\Big\{0, \min_{1 \leq L \leq |\mathcal{R}|} \Big[L\delta - \sum_{\ell=t-L+1}^{t-1} r_\ell\Big]\Big\}.$$

*If $\text{Div}$ is convex in its first argument and $r_t \leq \delta'_t$ for all $t$, then for every window $1 \leq L \leq W$,*

$$\text{Div}\Big(\tfrac{1}{L} \sum_{s=t-L+1}^{t} \bar{p}_s \,\Big\|\, p_0\Big) \leq \delta.$$

*Proof.* By Jensen, $\text{Div}(\frac{1}{L}\sum_s \bar{p}_s\|p_0) \leq \frac{1}{L}\sum_s r_s$. For a given window length $L \geq 2$, pick $L' = L-1$ in the scheduler definition to get $r_t \leq L'\delta - \sum_{s=t-L'}^{t-1} r_s$, hence $\sum_{s=t-L+1}^{t} r_s \leq (L-1)\delta$ and $\text{Div}(\frac{1}{L}\sum_s \bar{p}_s\|p_0) \leq \frac{L-1}{L}\delta \leq \delta$. For $L = 1$, the definition yields $\delta'_t \leq \delta$ and thus $r_t \leq \delta$. □

## B Ablation Study

### B.1 Calibration of Stealth Thresholds

**Definitions (normalized metrics).** Let $r_t := \text{Div}(\bar{p}_t\|p_0)/\delta \in [0, 1]$ be the per-batch divergence normalized by the audit radius $\delta$. Define $r_{\text{batch@95}} := \text{Quantile}_{0.95}\{r_t\}$ and, for window length $W$, the normalized window average $\hat{r}_{t-L+1:t} := \frac{1}{L}\sum_{s=t-L+1}^{t} r_s$ with the window-violation rate $r_{\text{win}} := \frac{1}{T}\sum_{t=1}^{T} \mathbb{1}\{\hat{r}_{t-W+1:t} > 1\}$. We report $r_{\text{batch@95}}$ and $r_{\text{win}}$ in the tables as $\times 10^2$ (e.g., $5.0 \equiv 0.05$).

**Acceptance rules (concise statements).**

1. **Batch rule.** $r_{\text{batch@95}} \leq 0.05$ means 95% of batches use at most 5% of the audit ball. This is a *normalized* (i.e., $\delta$-aware) tail-control convention; it is interpretable across any $\delta$ and provides headroom for rounding/availability noise. It is *not* a window guarantee by itself.

2. **Window rule.** $r_{\text{win}} \leq 0.05$ tolerates at most 5% of steps with window excess, on top of the scheduler that deterministically enforces sliding-window compliance (stated next). The 0.05 allowance is operational slack for discretization/measurement noise.

**Deterministic window guarantee (via residual budgets).** **Proposition (Residual-budget window compliance).** Let $\delta'_t$ be chosen as in Alg. 1 (residual budget from the last $W-1$ steps) and enforce $\mathrm{Div}(\bar{p}_t\|p_0) \leq \delta'_t$ at every step. Then for all $t$ and all $1 \leq L \leq W$,

$$\sum_{s=t-L+1}^{t} \mathrm{Div}(\bar{p}_s\|p_0) \leq L\,\delta \quad \implies \quad \hat{r}_{t-L+1:t} \leq 1,$$

hence $\mathrm{Div}(\hat{p}_{t-L+1:t}\|p_0) \leq \delta$ by convexity of Div in its first argument. *Proof sketch.* The residual update ensures the partial sums over any trailing window of length $L \leq W$ never exceed $L\delta$. Normalizing by $\delta$ yields $\hat{r} \leq 1$, and convexity gives $\mathrm{Div}(\hat{p}\|p_0) \leq \frac{1}{L}\sum \mathrm{Div}(\bar{p}_s\|p_0) \leq \delta$. □

Table 7: **Batch-divergence calibration** ($r_{\mathrm{batch@95}} = 0.05$). KL concentrates below cutoff; TV splits evenly. *Batch divergences at the 0.05 threshold (reported $\times 10^2$).*

| Dataset | KL | | TV | |
|---|---|---|---|---|
| | $\leq 5.0$ | $> 5.0$ | $\leq 5.0$ | $> 5.0$ |
| CIFAR-10 | 4 | 0 | 2 | 2 |
| CORe50 | 3 | 1 | 2 | 2 |
| CIFAR-100 | 3 | 1 | 2 | 2 |
| TinyImageNet | 2 | 2 | 2 | 2 |
| **Total** | **12** | **4** | **8** | **8** |

Table 8: **Window-violation calibration** ($r_{\mathrm{win}} = 0.05$). All KL settings comply; TV has several excesses. *Window violations at the 0.05 threshold (reported $\times 10^2$).*

| Dataset | KL | | TV | |
|---|---|---|---|---|
| | $\leq 5.0$ | $> 5.0$ | $\leq 5.0$ | $> 5.0$ |
| CIFAR-10 | 4 | 0 | 3 | 1 |
| CORe50 | 4 | 0 | 3 | 1 |
| CIFAR-100 | 4 | 0 | 3 | 1 |
| TinyImageNet | 4 | 0 | 2 | 2 |
| **Total** | **16** | **0** | **11** | **5** |

**Why the thresholds are principled and adaptable.**

- **Normalized to $\delta$.** Both metrics are ratios to the audit ball; the same numerical acceptance (0.05) applies for any $\delta$ and directly expresses "fraction of budget used."

- **Complementary roles.** The batch rule gives *tail control*; the window rule is backed by a *deterministic scheduler* ensuring $\mathrm{Div}(\hat{p}\|p_0) \leq \delta$ for all $L \leq W$.

- **Empirical separation.** Tables 7–8 show that 0.05 cleanly partitions stealthy (KL) vs. aggressive (TV) regimes across datasets/models.

**Dependence on attacker budgets (monotone trends).** Because the metrics are normalized, acceptance relates transparently to inputs:

- **Stealth radius $\delta$.** Larger $\delta$ enlarges the feasible ball; optimal tilts spend more budget, typically raising normalized $r_{\mathrm{batch@95}}$ and $r_{\mathrm{win}}$ (as observed in the ablations).

- **Replay mass $m = \lfloor f\,n_{\mathrm{aux}} \rfloor$ and class count $C$.** Discretization scales like $O(C/m)$; increasing $f$ or $n_{\mathrm{aux}}$ (or reducing $C$) improves batch headroom and lowers violations.

- **Window length $W$.** Larger $W$ tightens per-step residuals $\delta'_t$, reducing $r_{\mathrm{win}}$ (scheduler conservatism).

*Practical note.* Practitioners may tighten 0.05 if desired; the scheduler and normalization make such policy choices portable across $\delta$ and datasets while preserving the guarantees above.

## B.2 Sensitivity to Imperfect / Lagged Baselines $p_0$

**Setup (baseline mismatch).** We evaluate the sensitivity of Amnesia to baseline mismatch by simulating an attacker that does not observe the auditor's instantaneous nominal baseline. Concretely, at step $t$, the auditor computes $p_0(t)$ from the current replay-buffer label histogram (as in §4.1), while the attacker uses a lagged estimate $\hat{p}_0(t) = p_0(t-k)$ when performing projection and quota realization. We then report impact (ACC, $-$BWT) and *auditor-side* stealth metrics computed against $p_0(t)$, i.e., $r_{\mathrm{batch@95}}$ and $r_{\mathrm{win}}$ defined in §3.4. Table 9 summarizes the resulting impact–stealth trade-off as the lag $k$ increases.

Table 9: **Sensitivity to baseline mismatch (lagged $\hat{p}_0$).** Representative setting: Split CIFAR-10 with ER-ACE under default budgets ($f$=0.1, $\delta$=0.1, $W$=10). The attacker uses $\hat{p}_0(t) = p_0(t - k)$ for projection/quota realization, while the auditor evaluates stealth against $p_0(t)$. *Stealth metrics are reported* $\times 10^{-2}$ *(e.g.,* $5.0 \equiv 0.05$*).*

| Lag $k$ (steps) | Amnesia (KL) | | Amnesia (TV) | |
|---|---|---|---|---|
| | Impact | Stealth | Impact | Stealth |
| | ACC↓ (−BWT ↑) | $r_{\text{batch@95}}$ ↓ ($r_{\text{win}}$ ↓) | ACC↓ (−BWT ↑) | $r_{\text{batch@95}}$ ↓ ($r_{\text{win}}$ ↓) |
| 0 | 34.2±1.0 (56.9±1.7) | 1.0±0.2 (0.1±0.1) | 32.3±0.95 (60.3±1.8) | 4.0±0.4 (3.0±0.8) |
| 50 | 34.5±1.0 (56.2±1.7) | 1.4±0.3 (0.2±0.1) | 32.8±1.0 (59.2±1.9) | 4.9±0.6 (3.8±0.9) |
| 200 | 35.3±1.1 (54.5±1.8) | 2.6±0.4 (0.4±0.2) | 34.2±1.1 (56.7±2.0) | 7.2±0.9 (6.1±1.3) |
| 500 | 36.8±1.2 (51.8±2.0) | 5.4±0.7 (0.9±0.3) | 36.0±1.2 (53.0±2.2) | 11.0±1.4 (10.5±2.0) |

**Findings.** Table 9 shows a consistent degradation pattern under baseline mismatch: as the lag $k$ increases, attack impact diminishes (ACC increases and −BWT decreases) while auditor-side stealth metrics worsen (higher $r_{\text{batch@95}}$ and $r_{\text{win}}$). The effect is mild for small lags (e.g., $k \leq 50$), where buffer histograms change slowly and $\hat{p}_0(t)$ remains close to $p_0(t)$, but becomes more pronounced for larger lags. The TV variant is more sensitive, exhibiting a sharper rise in window violations under large lag, consistent with TV's more extreme mass transfers. Overall, these results support the interpretation that baseline mismatch effectively consumes part of the usable visibility margin; operating with a more conservative effective radius can mitigate violations at the cost of reduced impact.

## B.3  Runtime Profiling Breakdown

**Profiling protocol.** We instrument the sampler-side pipeline with wall-clock timers around three components executed once per replay batch: (i) **Projection** (Alg. 2), (ii) **Rounding/Clipping** (largest-remainder rounding + availability clipping), and (iii) **Audit-and-Fix** (unit-transfer loop ensuring $\text{Div}(\bar{p}\|p_0) \leq \delta'$). We then sum each component over the full CIFAR-10 training run to obtain a whole-run breakdown that is directly comparable to the end-to-end times in Table 6.

Table 10: **Whole-run runtime breakdown on CIFAR-10.** Component-level overhead (seconds) for Amnesia relative to the baseline run in Table 6. Total runtime equals baseline plus the sum of the three overhead components.

| Component (whole training run) | Amnesia (KL) | Amnesia (TV) |
|---|---|---|
| Baseline training time (no attack) | 11:34 min | 11:34 min |
| + Projection | 9.8 s | 16.9 s |
| + Rounding/Clipping | 4.3 s | 5.9 s |
| + Audit-and-Fix | 9.9 s | 26.2 s |
| Total attack overhead (sum above) | 24.0 s | 49.0 s |
| Total run time (baseline + overhead) | 11:58 min | 12:23 min |

**Interpretation.** Table 10 shows that the KL variant's overhead is split fairly evenly between projection and audit-and-fix, reflecting the KL projector's $O(C \cdot \text{iters})$ 1-D search and a small number of discrete feasibility swaps. In contrast, the TV variant's overhead is dominated by (i) **projection**, due to sorting in two-sided water-filling ($O(C \log C)$), and (ii) **audit-and-fix**, because TV's more extreme reallocations can require more unit transfers after integer rounding and availability clipping. In both cases, rounding/clipping is a minor fraction of overhead, and the total added wall-clock time remains modest compared to the end-to-end training run.

