# OpenReview forum: "Amnesia: A Stealthy Replay Attack on Continual Learning Dreams"
_TMLR — Accepted by TMLR_

### Review · Reviewer_c9iM · 2026-03-07

**Summary Of Contributions:**

The paper proposed an attack for CL that requires replay to retain the performance. The design of attack is interesting, instead of manipulating the data, the method modify the sampler to result in the change of replayed data during continuous learning. The proposed method is evaluated on multiple replay-based CL methods, and it successfully reduces the model utility when being hard to observe.

**Audience:**

Yes

**Audience Explanation:**

CL is an important topic in ML as the data is keep increasing every day, designing a model can conduct lifelong learning is important; therefore, learning what are possible way to attack the CL can potentially learn how to defense later.

**Claims And Evidence:**

Yes

**Claims Explanation:**

Section 3.4 further discuss its visibility and efficiency guarantee, demonstrating the effectiveness of the proposed method, and the experiments echo those guarantees.

**Requested Changes:**

1. As the current trend in CL is based on larger model and applying the rehearsal-free approaches, would this attack be applicable to those methods? E.g. OVOR [1], CODA-P [2]

References:
[1] OVOR: OnePrompt with Virtual Outlier Regularization for Rehearsal-Free Class-Incremental Learning
[2] CODA-Prompt: COntinual Decomposed Attention-based Prompting for Rehearsal-Free Continual Learning

2. Following the above, those methods do not finetune the whole model during the CL, how the effectiveness of the proposed attack over those parameter-efficient fine-tuning CL methods. Or more in general, other CL methods, e.g. generator as replay, prototype method, etc.

Others:
- The last sentence at page 12 is repeated. " not just the model,"
- The algorithm 1 and 2 under algorithm 2 should be 3 and 4.
- Fix cross reference in the description before sec 3.1. Please check the cross-ref in the paper.

---

> ### Author Response · Authors · 2026-03-17
> **Authors Response**
>
> **Comment:** We thank the reviewer for the thoughtful questions and helpful presentation feedback. We respond point-by-point below and indicate the concrete manuscript changes we made.
>
> > **(a) As the current trend in CL is based on larger model and applying the rehearsal-free approaches, would this attack be applicable to those methods? E.g. OVOR [1], CODA-P [2]**
>
> **Answer:** Thank you for this important question about how Amnesia relates to the current trend toward larger pretrained models and rehearsal-free continual learning. Amnesia’s threat model is explicitly scoped to replay-based CL pipelines where a separate replay sampler (or data service) selects buffer indices under auditable telemetry constraints (e.g., replay rate and replay-label histograms). OVOR and CODA-Prompt are rehearsal-free approaches that mitigate forgetting via prompting or parameter-efficient adaptation without sampling stored exemplars. Since Amnesia assumes the attacker can manipulate replay index selection, it is not directly applicable to rehearsal-free pipelines in their standard form; there is no replay buffer and no replay index set $I_t$ to control.
>
> That said, our broader systems takeaway still holds: whenever a CL pipeline introduces an explicit selection or routing mechanism (e.g., replay sampling, prompt routing, or mixture weights) that is only verified through lightweight logs, that component can become an attackable control point. Studying prompt-routing or composition attacks is a natural analogue, but it is outside the scope of the current replay-index threat model.
>
> **Manuscript modifications (what + where).**
>
> * **Section 3.2 (Threat Model & Attack Surface), opening paragraph:** we added one clarifying sentence stating that Amnesia assumes a replay buffer and an index-selection surface, and therefore does not directly apply to fully rehearsal-free methods (e.g., prompting methods without stored exemplars).
> * **Conclusion (Limitations paragraph):** we expanded the scope/limitations statement to explicitly distinguish:
>
>   * *Direct applicability:* exemplar replay with an index sampler
>   * *Non-applicability:* fully rehearsal-free CL without replay
>   * *Conceptual extensions:* other selection/composition modules (e.g., prompt routing), left as future work
>
> > **(b) Following the above, those methods do not finetune the whole model during the CL, how the effectiveness of the proposed attack over those parameter-efficient fine-tuning CL methods. Or more in general, other CL methods, e.g. generator as replay, prototype method, etc.**
>
> **Answer:** Amnesia is a sampler-level attack: it manipulates which past examples (or classes) are replayed under audited histogram and rate constraints. This mechanism is largely orthogonal to whether the learner fine-tunes the entire backbone or only a small PEFT module (e.g., adapters, LoRA, prompts, or the head). If replay is used, the replay distribution still determines the gradient signal delivered to whatever parameters are trainable, so the attack surface persists for replay-based PEFT CL.
>
> More broadly, we agree that other CL paradigms are important. Our current work directly targets stored-exemplar replay where an attacker controls the replay index sampler. However, the core concept, composition under auditable budgets, extends to any method that depends on sampling or weighting “past information” with controllable proportions:
>
> * **Generative replay:** the analogue is manipulating the class mixture of generated pseudo-samples under auditable constraints.
> * **Prototype methods:** the analogue is manipulating which prototypes are selected, or how often they are selected, again under auditable selection constraints.
>
> These settings are conceptually aligned, but they require a different integration point (generator or prototype selector rather than a buffer index sampler), and we therefore treat them as extensions beyond the present experimental scope.
>
> **Manuscript modifications (what + where).**
>
> * **Section 3.2 (Threat Model & Attack Surface):** we clarified that the prerequisite for Amnesia is replay and a controllable sampling interface, not full-model fine-tuning.
> * **Conclusion (Limitations paragraph):** we added a concise scope statement clarifying:
>
>   * *Replay-based index selection:* in scope (including replay-based PEFT CL)
>   * *Generative/prototype sampling:* conceptual extension and future work
>   * *Rehearsal-free prompting without replay:* out of scope
>
> > **Format / presentation comments**
>
> * **Conclusion:** removed the duplicated phrase (“not just the model,”) and corrected the associated typos.
> * **Algorithm naming/numbering:** corrected the internal labeling so the projection solvers are referenced consistently and no longer conflict with global algorithm numbering.
> * **Cross-references (before Sec. 3.1 and in Methodology):** fixed broken “Alg. ?? / Eq. ??”-style references and ensured all algorithm and table references compile correctly.

---

### Review · Reviewer_fVLf · 2026-03-10

**Summary Of Contributions:**

## Paper summary

In this paper, the authors introduce Amnesia, a new attack targeting the experience replay mechanism in Continual Learning (CL). One of the common solutions to alleviate catastrophic forgetting in CL is episodic reply where old samples are periodically used during the training. This new attack manipulates the selection of indices from the replay buffer. The attack is designed to be "stealthy" by adhering to auditable budgets: a visibility budget ($\delta$) bounding the divergence (TV or KL) from a nominal class histogram, and a mass budget ($f$) fixing the replay rate.

The goal in this attack is to cancel the impact of reply by maximizing forgetting. The authors achieve this goal with a two-step optimization (Tilt + Projection). Evaluation across Split CIFAR, CORe50, and Tiny-ImageNet against various baselines shows significant accuracy drops and increased forgetting (BWT).

## Contributions

* Novel Threat Model

* Mathematical Rigor

* Extensive empirical results

**Audience:**

Yes

**Audience Explanation:**

This paper is a strong contribution to the robustness of CL. While most of the existing work focus on new method for CL or simple attacks, this paper shifts the focus to "which" data is replayed, exposing a stealthy surface that is difficult to secure. This is an attempt for new types of attacks and shows the importance of data selection stage especially in episodic reply methods.

**Claims And Evidence:**

Yes

**Claims Explanation:**

The authors have properly explained all the important metrics in continual learning. Furthermore, they have illustrated their treat model and algorithm. Finally, the experiment section is very informative and comprehensive and shows the efficacy of the attack/threat model.

**Requested Changes:**

* Currently, the attack assumes the sampler knows exactly what the auditor considers "nominal." If the defender uses a moving-target $p_0$ (e.g., a randomized baseline), the projection might be misaligned.Action. It would be very beneficial to have a brief "Sensitivity Analysis" section or paragraph discussing how the attack performs if the attacker has an imperfect or lagged estimate of the auditor's $p_0$.

* The paper assumes a fixed number of classes $C$ for $p_0$. However in  "Class-Incremental" CL, $C$ grows. My question is that in the Methodology how the divergence constraints are calculated when the label space expands?

* The authors have already reported the end-to-end time, a more granular breakdown of the Projection Step vs. Audit-and-Fix Step would be beneficial, especially for the TV variant which requires sorting.

---

> ### Author Response · Authors · 2026-03-18
> **Authors Response**
>
> > **(2) The paper assumes a fixed number of classes $(C)$ for $(p_0)$. However in "Class-Incremental" CL, $(C)$ grows. My question is that in the Methodology how the divergence constraints are calculated when the label space expands?**
>
> **Answer:** Thank you for this clear question; this is an important notation and definition point in class-incremental learning. We used a fixed $(C)$ primarily for notational simplicity. In class-incremental CL, at step $(t)$ we operate on the current seen label space of size $(C_t)$: $(p_0)$, $(p^\star)$, and $(\bar p_t)$ are elements of $(\Delta^{C_t})$. When new classes appear, we append the new coordinates and recompute $(p_0)$ from the current buffer histogram; the divergence audit $(\mathrm{Div}(\bar p_t \mid p_0))$ is then evaluated in this expanded space. For KL, we apply standard smoothing so that $(p_{0,c} > 0)$, keeping $(\mathrm{KL}(\cdot \mid p_0))$ finite. Importantly, our main benchmarks are class-incremental, so the reported experiments already reflect the expanding-label setting.
>
> **Manuscript modifications (what + where).**
>
> * **Section 3.1 (Preliminaries & Notation), first paragraph:** we updated the notation from $(C)$ to $(C_t)$ and stated explicitly that histograms live in $(\Delta^{C_t})$ in class-incremental CL.
> * **Section 3.3 (Step C: quotas), rounding bound sentence:** we replaced the bound $(|\bar p - p^\star|_1 \le C/m)$ with $(|\bar p - p^\star|_1 \le C_t/m)$ to reflect the current seen label space.

---

> ### Author Response · Authors · 2026-03-18
> **Authors Response**
>
> > **(3) The authors have already reported the end-to-end time, a more granular breakdown of the Projection Step vs. Audit-and-Fix Step would be beneficial, especially for the TV variant which requires sorting.**
>
> **Answer:** Thank you for requesting a more granular runtime breakdown, especially since TV projection requires sorting and audit-and-fix can involve swaps. We agree that end-to-end runtime alone does not reveal where the overhead arises. Our method has clear complexity drivers: KL projection is $(O(C \cdot \mathrm{iters}))$ via a one-dimensional search with $(O(C))$ evaluations; TV projection is $(O(C \log C))$ due to sorting in two-sided water-filling; rounding/clipping and audit-and-fix are $(O(C))$ plus a bounded number of unit transfers, with strong termination structure shown in the supplement. To make this transparent, we now provide a whole-run profiling breakdown of the overhead into the key components.
>
> **Manuscript modifications (what + where).**
>
> * **Main paper, runtime discussion near Table 6** we added a pointer sentence directing readers to the appendix for the component-level breakdown.
> * **Appendix, new “Runtime Profiling Breakdown” subsection:** we added a profiling table decomposing the full-run overhead into **(i)** Projection, **(ii)** Rounding/Clipping, and **(iii)** Audit-and-Fix for both KL and TV variants, along with a short interpretation paragraph.

---

### Review · Reviewer_uHVq · 2026-03-24

**Summary Of Contributions:**

This paper focuses on poisoning attacks for replay-based continual learning models by attacking the model’s sampler that samples from the replay buffer, while minimizing detectability by minimizing the deviation from the nominal class distribution computed using either KL or TV deviation. The method was evaluated against different continual learning methods primarily relying on experience replay.

**Audience:**

Yes

**Audience Explanation:**

Poisoning of continual learning models, specifically with the recent rise of language models, would be a critical direction for research. While the threat model might not be directly transferrable, the findings of this paper might turn out to be a useful stepping stone for further work.

**Claims And Evidence:**

Yes

**Claims Explanation:**

The authors provided theoretical arguments as well as experimental validation on four different datasets as well as three different model architectures in order to support the effectiveness of their approach against basic variations of replay-based continual learning methods (ER, ER-ACE, SCR, DER++). They also compared the effectiveness of their attack compared to three other attacks proposed in the past. While I am not fully convinced that the threat model is realistic given the assumption to control the sampler as well as the lagged metrics (gray-box threat model), the claims are still well justified within the described framework.

**Requested Changes:**

- While the paper assumes replay-based continual learning methods, can you comment on how the findings in this paper could be relevant for non-replay-based continual learning methods? While replay-based methods were dominant in the past, this might change in the future given the rise of continual learning for language models.
- Can you please fix the broken algorithm references in the paper (page # 5, 6, 10, and 11)?
- Can you please switch to \citep instead of directly using \cite which would separate out the citations appropriately?
- Can you please fix the use of "Eq. equation" instead of using either "Equation" or "Eq."?

---

> ### Author Response · Authors · 2026-03-25
> **Authors Response**
>
> > **(a) While the paper assumes replay-based continual learning methods, can you comment on how the findings in this paper could be relevant for non-replay-based continual learning methods? While replay-based methods were dominant in the past, this might change in the future given the rise of continual learning for language models.**
>
> **Answer:** Thank you for your insightful comment. Amnesia’s threat model is intentionally scoped to replay-based continual learning systems with a stored buffer and a controllable replay sampler, so it does not directly apply to fully rehearsal-free methods in their standard form. That said, we believe the broader systems takeaway extends beyond exemplar replay. The core vulnerability we highlight is not replay per se, but a low-privilege mechanism that controls which past information influences the update while being monitored only through lightweight telemetry. In replay-based vision CL this mechanism is the replay sampler; in future non-replay or language-model CL settings, analogous control points may include retrieval or memory selection, prompt/example selection, expert routing, or composition of parameter-efficient modules. We do not claim that Amnesia directly transfers to these settings as-is, but our results suggest a more general design lesson: modules that shape access to past information should be treated as first-class attack surfaces and audited accordingly.
>
> **Manuscript modifications (what + where).**
>
> * **Section 3.2 (Threat Model & Attack Surface):** we expanded the scope discussion to distinguish between (i) replay-based methods, which are directly in scope, and (ii) fully rehearsal-free methods, which are not directly attacked by Amnesia in its current form, while explicitly noting conceptual analogues for language-model continual learning.
> * **Conclusion (Limitations / Future Work):** we added a sentence clarifying that the broader lesson of the paper may extend to other “past-information selection” mechanisms such as retrieval, prompt/example selection, or routing/composition, which we left to future work.
>
>
> ---
>
> ### Comment:
>
> **“While I am not fully convinced that the threat model is realistic given the assumption to control the sampler as well as the lagged metrics (gray-box threat model), the claims are still well justified within the described framework.”**
>
> **Response:**
> Thank you for this careful and fair assessment. We agree that the threat model is intentionally scoped and may not match every deployment setting. Our goal in this paper was not to claim universal realism, but rather to isolate and study a concrete, low-privilege attack surface that can plausibly arise in replay-based CL pipelines when replay index selection is handled by a separate sampler/service.
>
> A small clarification may also help here: in our threat model, the attacker controls the **sampler/index selection**, but does **not** control the lagged metrics themselves. Those metrics are assumed to be read-only, asynchronously logged telemetry or metadata produced by the training system, which the sampler can access but not modify. We designed the threat model this way specifically to separate sampler-side interference from stronger compromises involving model parameters, gradients, losses, or data corruption.
>
> ---
>
> ### Comment:
>
> **“While the threat model might not be directly transferrable, the findings of this paper might turn out to be a useful stepping stone for further work.”**
>
> **Response:**
> Thank you,  we appreciate this perspective, and it aligns well with how we view the contribution of the paper. Our primary goal is not to argue that the exact threat model transfers unchanged to all continual-learning settings, but to establish a principled starting point for reasoning about security vulnerabilities in replay/data-selection mechanisms.
>
> More broadly, we hope the paper helps motivate follow-up work in several directions: weaker or alternative attacker models, stronger auditing and defenses, and analogous attack surfaces in other forms of continual learning where some mechanism controls access to past information. In that sense, we are encouraged by the reviewer’s interpretation of the work as a stepping stone, because that is very much the role we hope it will play.

---

> ### Author Response · Authors · 2026-03-25
> **Authors Response**
>
> > **(b) Can you please fix the broken algorithm references in the paper (page # 5, 6, 10, and 11)?**
>
> **Answer:** Thank you for flagging these broken algorithm references. We agree that the current use of pseudo-subalgorithm notation inside a single float (e.g., Alg. 2.1 / Alg. 2.2) is fragile and can render inconsistently. In the revision, we replaced these with stable named references to the KL and TV solvers within the shared projection-solver figure/algorithm, and we ensured that all cross-references compile correctly throughout the main text.
>
> **Manuscript modifications (what + where).**
>
> * **Section 3.3 (Methodology):** we fixed all references to the projection solvers in the paragraph introducing the tilt-then-project step, in Algorithm 1 comments, and in the KL/TV bullet descriptions.
> * **Section 5 (Results):** we fixed the later references in the KL-vs-TV discussion so they point consistently to the named solver references rather than pseudo-numbered subalgorithms.
>
> > **(c) Can you please switch to `\citep` instead of directly using `\cite` which would separate out the citations appropriately?**
>
> **Answer:** Thank you for pointing this out. We agree. In the current draft, the citations are used parenthetically, so switching from `\cite` to `\citep` is the appropriate style choice and also improves the formatting of grouped citations.
>
> **Manuscript modifications (what + where).**
>
> * **Global citation cleanup:** we replaced parenthetical uses of `\cite{...}` with `\citep{...}` throughout the main paper and appendix.
>
> > **(d) Can you please fix the use of “Eq. equation” instead of using either “Equation” or “Eq.”?**
>
> **Answer:** Thank you for catching this formatting issue. We agree. This arose from combining Eq. with `\eqref{...}` under the current style, which rendered as Eq. equation N. We corrected these references so that they consistently render as either Eq. N or Equation N; in the revision we used the shorter `Eq.~\ref{...}` form for consistency with the rest of the paper.
>
> **Manuscript modifications (what + where).**
>
> * **Section 3.3 and Section 3.4:** we replaced the affected equation references around Eq. 2 and Eq. 4 with a consistent `Eq.~\ref{...}` style.

---

> ### Comment · Reviewer_uHVq · 2026-04-01
> **Thank you for the responses**
>
> I'm thankful to the authors for responding to my questions/comments. I'm happy to see that the authors expanded the discussion around rehearsal-free attacks in the paper as this might potentially be the next big focus. Everything else is aligned with what I would expect and understand from the paper, but appreciate the clarifications from the authors. As the authors mentioned, I think the main contribution is to focus on low-privileged attacks, even if the particular threat model might be contrived to make it feasible to study. I also appreciate all the formatting fixes to the paper.

---

### Author Response · Authors · 2026-03-25
**Summary of changes in the revised manuscript**

We sincerely thank the reviewers for their thoughtful and constructive feedback. The reviews primarily focused on clarifying the scope of the threat model, its applicability beyond replay-based continual learning, assumptions about the audit baseline and class-incremental notation, runtime transparency, and several presentation issues. In our rebuttal, we address all major concerns and specify the corresponding manuscript revisions as follows:

- **Reviewer c9iM(a) + Reviewer uHVq(a) - Applicability beyond replay-based continual learning.** We clarify that Amnesia is explicitly scoped to replay-based continual learning systems with a stored buffer and a controllable replay sampler, and therefore does not directly apply to fully rehearsal-free methods in their standard form. At the same time, we now emphasize the broader systems lesson of the paper: any module that controls access to past information under lightweight auditing - such as retrieval, prompt/example selection, expert routing, or module composition - can become a comparable attack surface. These clarifications are incorporated in **Section 3.2 (Threat Model & Attack Surface)** and expanded in the **Conclusion / Limitations**.

- **Reviewer c9iM(b) - PEFT and other continual-learning paradigms.** We clarify that Amnesia is a sampler-level attack and is therefore largely orthogonal to whether the learner uses full fine-tuning or parameter-efficient adaptation. When replay is used, manipulating the replay distribution still changes the gradient signal delivered to the trainable parameters, so replay-based PEFT settings remain in scope. We further clarify that **generative replay** and **prototype-based methods** are conceptually related extensions, but require different control points and are therefore left as future work. These scope clarifications are added to **Section 3.2** and the **Conclusion / Limitations**.

- **Reviewer fVLf(1) - Nominal baseline $p_0$, moving-target audits, and sensitivity analysis.** We clarify that, in the default setting, $p_0$ is computed from sampler-visible telemetry, such as the current replay-buffer histogram or a moving average, and is therefore not hidden from the sampler. We also explain how the method behaves under lagged, noisy, or partially hidden estimates of $p_0$, where more conservative stealth margins are required. To make this explicit, we add a new **appendix sensitivity analysis** and corresponding table that vary the lag between $\hat p_0$ and $p_0$ and report the resulting impact - stealth trade-off. The threat-model clarification appears in **Section 3.2**, with the new ablation placed in the **Appendix**.

- **Reviewer fVLf(2) - Expanding label space in class-incremental learning.** We revise the notation to reflect the fact that the effective label space grows over time in class-incremental settings. Specifically, we replace $C$ with $C_t$, clarify that $p_0$, $p^\star$, and $\bar p_t$ lie in $\Delta^{C_t}$, and update the rounding bound accordingly from $C/m$ to $C_t/m$. These notation corrections are made in **Section 3.1 (Preliminaries & Notation)** and **Section 3.3 (Step C: quotas)**.

- **Reviewer fVLf(3) - Runtime breakdown and implementation overhead.** We supplement the end-to-end runtime discussion with a finer-grained profiling breakdown that separates the overhead into **Projection**, **Rounding/Clipping**, and **Audit-and-Fix** for both the **KL** and **TV** variants. The main paper now points readers to a new **Appendix subsection on Runtime Profiling Breakdown**, where the profiling table and interpretation are provided.

- **Reviewer c9iM + Reviewer uHVq - Presentation, formatting, and typographical corrections.** We consolidate all presentation-related issues into a comprehensive cleanup of the manuscript. This includes removing duplicated phrasing and typos in the Conclusion, fixing broken algorithm, equation, and table cross-references, replacing fragile pseudo-subalgorithm numbering with stable named references for the **KL** and **TV** solvers, switching parenthetical citations from `\cite{...}` to `\citep{...}`, and standardizing equation references to a consistent `Eq.~\ref{...}` style. These fixes affect **Section 3.3**, **Section 3.4**, **Section 5**, the **Conclusion**, and the **Appendix** as needed.

Overall, these changes substantially improve the clarity, scope definition, and presentation quality of the manuscript. We believe the revised version now makes the threat model more precise, strengthens the discussion of applicability and limitations, provides better transparency regarding runtime and robustness, and resolves the formatting issues identified by the reviewers. We sincerely thank the reviewers again for their constructive feedback and believe these revisions make the paper significantly stronger.

---

### Decision · Action_Editor_PCc6 · 2026-05-14

**Recommendation:** Accept as is

**Additional Comments:**

The paper proposes a stealthy attack against continual learning systems that use experience replay, where the attacker targets the replay mechanism (the process with which samples are selected from the replay buffer). It is stealthy by minimizing the deviation from the "nominal" class distribution.

The key concerns that were raised in the reviewers are about the transferability of the threat model and findings presented in this work to modern continual learning algorithms that do not use experience replay, to larger models and different learning locations (all parameters vs PEFT). The authors have clarified that, while their specific attack is specifically designed to target the replay mechanism, the key lessons from this work may also apply to other modules that control access to past information under lightweight auditing (such as retrieval, expert routing, or other aspects of modern systems). The general idea of considering such modules as attack surfaces is interesting and may inspire future work to investigate this issue more broadly, in the context of different continual learning systems. Other clarifications and writing improvements made during the rebuttal seem to address the reviewers' comments sufficiently.

**Audience:**

Yes

**Audience Explanation:**

Reviewers agreed that continual learning is an important topic and understanding vulnerabilities of these systems is of interest to the community. Some reviewers found that the threat model used in this work isn't realistic, and that there is a recent trend away from replay-based algorithms, possibly lessening the contribution of this work if it's not directly applicable to modern pipelines. However, as the authors also pointed out, and reviewers acknowledged, some lessons may transfer to other types of algorithms (the authors have explicitly discussed analogies in their rebuttal), making the contributions of this work an interesting stepping stone for studying this problem for a broader set of methods.

**Claims And Evidence:**

Yes

**Claims Explanation:**

The paper proposes a stealthy attack against continual learning systems that use experience replay, where the attacker targets the replay mechanism (the process with which samples are selected from the replay buffer). It is stealthy by minimizing the deviation from the "nominal" class distribution.

The reviewers all agree that the claims made in the submission are supported by accurate, convincing and clear evidence. Specifically, the reviewers noted that the authors provide both theoretical arguments as well as experimental validation to support their claims. The reviewers found the experimental investigation to be "very informative and comprehensive". The experiments are conducted on four different datasets and three different model architectures. They also compare the effectiveness of their attack against previously-proposed attacks.